# All Roads Lead to Likelihood: The Value of Reinforcement Learning in Fine-Tuning

**Gokul Swamy**
Carnegie Mellon University

**Sanjiban Choudhury & Wen Sun**
Cornell University

**Zhiwei Steven Wu & J. Andrew Bagnell**
Carnegie Mellon University

## Abstract

From a first-principles perspective, it may seem odd that the strongest results in foundation model fine-tuning (FT) are achieved via a relatively complex, two-stage training procedure. Specifically, one first trains a reward model (RM) on some dataset (e.g., human preferences) before using it to provide *online* feedback as part of a downstream reinforcement learning (RL) procedure, rather than directly optimizing the policy parameters on said dataset via *offline* maximum likelihood estimation. In fact, from an information-theoretic perspective, we can only *lose* information via passing through a reward model and cannot create any new information via on-policy sampling. To explain this discrepancy, we scrutinize several hypotheses on the value of RL in FT through both theoretical and empirical lenses. Of the hypotheses considered, we find the most support for the explanation that on problems with a *generation-verification gap*, *(1)* it is relatively easy to learn the relatively simple RM (*verifier*) from the preference data. Then, *(2)* the downstream RL procedure only returns policies (*generators*) that are optimal for such relatively simple verifiers. Thus, end-to-end, two-stage online FT only has to search over a reduced subset of the full space of policies, requiring less data than offline FT.

## 1 Introduction

Whether one refers to it as reinforcement learning from human feedback (RLHF, Christiano et al. (2017)), preference fine-tuning (PFT), or even "alignment," the last step in the training pipeline of a wide variety of foundation models (FMs) is fundamentally concerned with raising the generation likelihood of preferred completions of a prompt relative to those of dis-preferred completions.

From this perspective, a natural question may be why anything other than maximum likelihood estimation (MLE) – i.e., standard supervised learning – is needed for the PFT problem. Indeed, a plethora of *offline* approaches to PFT that directly optimize policy parameters via solving a (regularized) classification problem on preference data have been proposed in the literature (e.g., DPO (Rafailov et al., 2023), IPO (Azar et al., 2023), SLiC-HF (Zhao et al., 2023)).

However, when one looks at the training procedure of today's most capable models (Achiam et al., 2023; Team et al., 2024; Dubey et al., 2024), one almost always sees a relatively complex two-stage procedure adopted instead. First, one learns a reward model (RM) – i.e., a classifier – on the preference data, before using it to provide labels for a downstream *online* reinforcement learning (RL) procedure that ultimately optimizes the policy's parameters (Bai et al., 2022; Ouyang et al., 2022; Stiennon et al., 2020; Gao et al., 2024a;b; Calandriello et al., 2024; Guo et al., 2024).

Across academic (Tajwar et al., 2024; Xu et al., 2024; Song et al., 2024a), industry (Tang et al., 2024), and open-source (HuggingFace, 2024) comparisons, we robustly see the relatively complex, two-stage online techniques out-perform simpler purely offline approaches. More generally, online approaches to supervised fine-tuning (SFT) have also been shown to out-perform vanilla next-token prediction (Sun & van der Schaar, 2024; Chen et al., 2024; Wulfmeier et al., 2024; Choudhury, 2025). Furthermore, recent models capable of complex reasoning (e.g., OpenAI's o1 (Jaech et al., 2024) or DeepSeek's r1 (Guo et al., 2025)) are still trained via on-policy RL rather than by using an offline

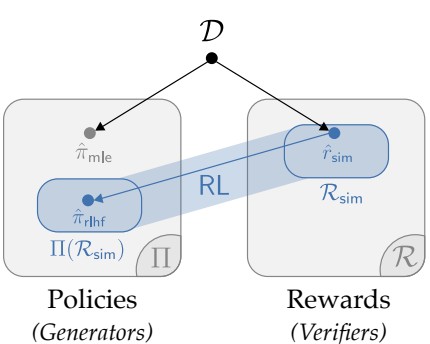

Policies
*(Generators)*

Rewards
*(Verifiers)*

Figure 1: While offline preference fine-tuning (PFT) procedures (e.g., DPO (Rafailov et al., 2023)) directly optimize policy parameters via (regularized) maximum likelihood estimation (MLE, i.e., $\hat{\pi}_{\mathsf{mle}} \in \Pi$), online PFT procedures (e.g., standard RLHF (Christiano et al., 2017)) first fit a reward model (RM) via MLE before using the RM to provide online feedback for a downstream reinforcement learning (RL) procedure. Empirically, this latter two-stage procedure often works better, despite the loss of information caused by passing through an RM. We find evidence that for problems with a *generation-verification gap*, this result may be best explained by viewing the first stage as finding a relatively simple *verifier* / RM $\hat{r}_{\mathsf{sim}} \in \mathcal{R}_{\mathsf{sim}}$, with the end-to-end search over *generators* / policies now simplified to just the subset of policies that are optimal for relatively simple verifiers (i.e., $\hat{\pi}_{\mathsf{rlhf}} \in \Pi(\mathcal{R}_{\mathsf{sim}}) \subset \Pi$).

MLE procedure, with academic investigations concurring (Chu et al., 2025; Setlur et al., 2025). Thus, we ask the following question:

$\mathbb{Q}$*: What is the value of two-stage, online FT if we just want to maximize data likelihood?*

Part of the challenge of providing a satisfactory answer to this question lies in the difficulty of applying traditional arguments about the value of online RL to FM post-training. First, the standard justification for interaction – the learner being able to observe and therefore learn to recover from their mistakes (Ross et al., 2011; Swamy et al., 2021) – does not appear to be true prima facie unless we somehow give a language model the ability to delete tokens (Cundy & Ermon, 2023; Wulfmeier et al., 2024). Second, the fact that it is common practice to use reward models that are at least as large as the policy makes it difficult to naively apply classical arguments predicated on the relative simplicity of rewards when compared to policies. This is because such arguments often implicitly assume one is learning a reward model over a relatively simple function class (e.g., Ng et al. (2000)).

Perhaps even more fundamentally, the *data processing inequality* (MacKay, 2003) tells us that we can only *lose* information when passing from the raw source of data to the reward model, and cannot create any information during on-policy sampling. Taken together, these points make it seem as though the "yellow-brick road" of online PFT may have been paved with pyrite rather than gold.

In response, we scrutinize a variety of hypotheses about the value of RL in PFT through theoretical and experimental lenses. We mostly ground our study in preference FT, but note that analogous arguments can be made for SFT and verifier-based RL settings. Our contributions are three-fold:

**1. We prove that under idealized assumptions, online and offline PFT techniques should return policies of equivalent quality.** Using tools from information geometry, we show that regardless of the coverage of preference dataset samples, offline and online PFT techniques have the same set of optima when the same function class is used for both policies and reward models.

**2. We provide evidence against several prior / novel hypotheses for the value of RL in PFT.** In particular, we provide evidence against explanations that rest solely on online techniques performing better regularization to the reference policy, a computational benefit provided by on-policy sampling, or the ability to use a wider distribution of data to train a reward model than to train a policy. While it is hard to conclusively rule out these factors, we provide evidence that they are not the whole story.

**3. We provide theoretical and empirical evidence for an alternative hypothesis on problems with a generation-verification gap.** Many problems in computer science are widely conjectured to have simpler *verifiers* / reward models than *generators* / optimal policies (Godel, 1956; Cook, 2023). We hypothesize that for such problems, it is easier to learn the relatively simple reward model than the relatively complex optimal policy from the preference data. RL then merely serves as a way to compute a (soft) optimal policy for this simple verifier. However, end-to-end, online PFT only has to search over just those policies that are optimal for relatively simple verifiers. In short, we argue that:

$\mathbb{A}$*: The value of two-stage interactive FT is derived from an end-to-end reduction the space of policies to search over to just those that are optimal for relatively simple verifiers.*

In the language of statistical learning, this hypothesis states that the real benefit of RL in fine-tuning is that it is the most convenient method that we know of for performing *proper learning* (Valiant, 1984), compared to *improper* offline FT. We find the least evidence against this hypothesis compared to all others we consider in this paper, which in some sense is the best we can hope for (Popper, 2014).

## 2 ON THE INFORMATION GEOMETRY OF FT

We begin with notation before deriving our core results. All proofs are located in Appendix B.

### 2.1 PRELIMINARIES

We consider a finite-horizon, reward-free Markov Decision Process (MDP, Puterman (2014)). Let $\mathcal{X}$ denote the set of initial states (i.e., prompts) and $\rho_0$ their distribution. We use $\mathcal{A}$ to denote the action space (i.e., set of tokens) and $\mathcal{S}$ to denote the state space (i.e., set of partial generations). The dynamics of our MDP are deterministic, known, and tree-structured: $\mathcal{T}(s'|s, a) = 1$ if $s' = s \circ a$ and 0 otherwise (i.e., we can only append tokens). We use $H$ to denote the horizon of our MDP (i.e., maximum generation length). A policy $\pi : \mathcal{S} \to \Delta(\mathcal{A})$ maps from a prefix to a distribution over next tokens. A trajectory (i.e., a prompt and its completion) $\xi$ is generated by sampling an initial state $s_0 \sim \rho_0$ and then sampling from the policy $H$ times. We use $\mathbb{P}_\pi(\xi)$ to denote the probability of sampling a trajectory $\xi$ under policy $\pi$ and use $\mathbb{P}_\Pi$ to denote the set of all such distributions over trajectories / generations induced by any $\pi \in \Pi \subseteq \{\mathcal{S} \to \Delta(\mathcal{A})\}$. We use $\xi \sim \pi$ as shorthand for sampling from such a distribution (i.e., sampling a generation from the policy). We use $\mathcal{D} = \{(\xi_i^+, \xi_i^-)\}_{i=1}^N$ to denote a dataset of trajectory-level preferences over pairs with the same $s_0$ and $\mathbb{P}_\mathcal{D}(\xi_1 \succ \xi_2|s_0)$ to denote the empirical probability that $\xi_1$ is preferred to $\xi_2$. The full space of trajectories is denoted by $\Xi$, and $\Xi|s_{0:h}$ is used to denote the set of trajectories with some prefix $s_{0:h}$. We use $\pi_{\text{ref}} \in \Pi$ to denote the reference policy to stay close to. Both $\arg\max$s and $\arg\min$s return the full set of optima.

### 2.2 GLOBAL AND LOCAL REWARD MODELS

Central to our study will the relationship between policies and trajectory-level reward models. We will use $\Pi$ to denote the set of policies and $\mathcal{R}$ to denote the set of reward models, with $r : \Xi \to \mathbb{R}$ for all $r \in \mathcal{R}$. We note that it is standard practice to use the same architecture (and often the same initial checkpoint and dataset) to train both policies and reward models. Specifically, reward models are often constructed by removing the final softmax at the end of the transformer (Vaswani et al., 2017) that gives us a distribution over tokens and replacing it with a single layer or a shallow MLP that eventually outputs a single scalar value (Rafailov et al., 2023). These models are then evaluated with the concatenated prompt and entire completion $s_H$ being passed into the transformer. Throughout this paper, we will use the term ***global*** to refer to such trajectory-level (i.e., non-Markovian) RMs.

Beyond merely having a shared backbone, a more precise isomorphism holds for reward models that take the form of the sum of policy log probabilities over the tokens in a generation (Degrave et al., 2019; Rafailov et al., 2023). More formally, we denote the set of ***local*** RMs $r_\pi : \Xi \to \mathbb{R}$ as

$$\mathcal{R}(\Pi) = \left\{ r_\pi(\xi) = \sum_{h=0}^H \log \pi(a_h|s_h) \middle| \pi \in \Pi \right\}. \tag{1}$$

It directly follows that $\mathcal{R}(\Pi)$ is isomorphic to $\Pi$.

### 2.3 A UNIFIED OBJECTIVE FOR FINE-TUNING

We now formulate a general loss function that both online and offline (P)FT methods optimize, albeit with different procedures. Loosely speaking, a variety of fine-tuning tasks (e.g., SFT, PFT) roughly fit within the template of the following reverse KL-regularized policy optimization problem:

$$\pi^\star = \arg\min_{\pi \in \Pi} \underbrace{\mathbb{E}_{z \sim \mathcal{D}} \left[ \mathbb{D}_{\text{KL}}(\mathbb{P}_\mathcal{D}(z)||\mathbb{P}_\pi^\mathcal{Z}(z)) \right]}_{\text{Data Likelihood}} + \underbrace{\beta \, \mathbb{D}_{\text{KL}}(\mathbb{P}_\pi(\xi)||\mathbb{P}_{\pi_{\text{ref}}}(\xi))}_{\text{Prior Reg.}}, \tag{2}$$

where $z \in \mathcal{Z}$ denotes an ordered pair of trajectories for PFT. The first *forward* KL term measures how likely samples from $\mathcal{D}$ are under the learned policy $\pi$ and the second *reverse* KL term keeps the

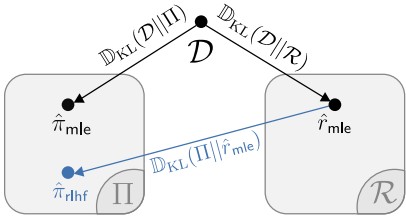

Figure 2: We can view both offline and offline PFT as solving Eq. 3. Offline methods directly project the preference dataset $\mathcal{D}$ onto policy class $\Pi$ under forward KL (§2.4), while online methods first project from $\mathcal{D}$ onto reward class $\mathcal{R}$ under forward KL (§2.4), before projecting onto $\Pi$ under reverse KL (§2.5). A similar characterization of online PFT is made in the concurrent work of Xiao et al. (2025).

completion probabilities of $\pi$ to be close to those of the reference policy $\pi_{\text{ref}}$ on-policy. Intuitively, if $\mathcal{D}$ had full coverage over all possible $z$, we would have no need for the second term, but due to finite-sample limitations, we add in a regularizer to prevent us from being lead too far astray (Gao et al., 2023; Song et al., 2024a). That said, for simplicity of presentation, we will consider the case where $\beta = 1$ and temporarily replace the second KL regularization term with entropy regularization:

$$\pi^\star = \underset{\pi \in \Pi}{\arg\min}\, \mathbb{E}_{z \sim \mathcal{D}} \left[ \mathbb{D}_{\text{KL}}(\mathbb{P}_{\mathcal{D}}(z) || \mathbb{P}_\pi^{\mathcal{Z}}(z)) \right] - \mathbb{H}(\pi), \tag{3}$$

where $\mathbb{H}(\pi) = \mathbb{E}_{\xi \sim \pi} \left[ \sum_h^H - \log \pi(a_h | s_h) \right]$ is the (causal) entropy of the policy (Ziebart, 2010).

## 2.4 Maximum Likelihood in PFT

We will first focus on minimizing the *forward* KL (FKL) term of (3), which is equivalent to maximum likelihood estimation (MLE). Given this MLE objective, both online and offline methods aim to find a model that best explains the data, but search over different model classes. The first stage of online PFT methods is MLE over reward models in $\mathcal{R}$. A common parametric model that relates rewards to preference probabilities is the Bradley-Terry (BT) Model (Bradley & Terry, 1952):

$$\mathbb{P}_r^{\text{BT}}(\xi_1 \succ \xi_2 | s_0) = \sigma\left(r(\xi_1) - r(\xi_2)\right), \tag{4}$$

where $\succ$ means preferred to and $\sigma$ is the sigmoid function. We can then maximize likelihood over global reward models, each of which parametrizes a probabilistic model via BT:

$$\hat{r}_{\text{mle}} = \underset{r \in \mathcal{R}}{\arg\min}\, \mathbb{E}_{(\xi_1, \xi_2) \sim \mathcal{D}} \left[ \mathbb{D}_{\text{KL}}(\mathbb{P}_{\mathcal{D}}(\xi_1 \succ \xi_2 | s_0) || \mathbb{P}_r^{\text{BT}}(\xi_1 \succ \xi_2 | s_0)) \right] \tag{5}$$

$$= \underset{r \in \mathcal{R}}{\arg\max} \sum_i^N \log \sigma(r(\xi_i^+) - r(\xi_i^-)). \tag{6}$$

Observe that fitting a global RM is essentially solving standard logistic regression or classification problem. In contrast, offline PFT methods instead perform MLE directly over the policy class $\Pi$. Recall that any policy $\pi \in \Pi$ is also a local RM $r_\pi \in \mathcal{R}(\Pi)$ (defined in (1)). Thus, we can fit a policy via MLE by substituting in the sum of log probabilities for $r_\pi$ in the Bradley-Terry likelihood:

$$\hat{\pi}_{\text{mle}} = \underset{\pi \in \Pi}{\arg\min}\, \mathbb{E}_{(\xi_1, \xi_2) \sim \mathcal{D}} \left[ \mathbb{D}_{\text{KL}}(\mathbb{P}_{\mathcal{D}}(\xi_1 \succ \xi_2 | s_0) || \mathbb{P}_{r_\pi}^{\text{BT}}(\xi_1 \succ \xi_2 | s_0)) \right] \tag{7}$$

$$= \underset{\pi \in \Pi}{\arg\max} \sum_i^N \log \sigma(r_\pi(\xi_i^+) - r_\pi(\xi_i^-)) \tag{8}$$

$$= \underset{\pi \in \Pi}{\arg\max} \sum_i^N \log \sigma \left( \sum_h^H \log \frac{\pi(a_{h,i}^+ | s_{h,i}^+)}{\pi(a_{h,i}^- | s_{h,i}^-)} \right) = \underset{\pi \in \Pi}{\arg\max}\, \ell_{\text{mle}}(\pi). \tag{9}$$

Ignoring the reference policy for a moment, this is what offline PFT approaches like DPO are at heart: a trajectory-level classification problem over local (rather than global) reward models.

## 2.5 Maximum Entropy in PFT

The second stage of online PFT corresponds to optimizing a learned reward and the second entropy term in (3). Given a learned global reward model $r$, one computes its soft-optimal policy $\pi_r^\star$, i.e.

$$\pi_r^\star = \underset{\pi \in \Pi}{\arg\max}\, \mathbb{E}_{\xi \sim \pi}[r(\xi)] + \mathbb{H}(\pi). \tag{10}$$

If we solved the above equation in closed-form over *all* policies (i.e., not just those in $\Pi$), we would compute a policy with prompt-conditioned trajectory (i.e., completion) distribution

$$\mathbb{P}_r^\star(\xi|s_0) = \frac{\exp(r(\xi))}{\sum_{\xi' \in \Xi|s_0} \exp(r(\xi'))} = \frac{\exp(r(\xi))}{Z(r, s_0)}, \tag{11}$$

as first proved by Ziebart (2010). Observe that for trajectories $\xi_1, \xi_2$ with shared prompt $s_0$,

$$\mathbb{P}_r^{\text{BT}}(\xi_1 \succ \xi_2|s_0) = \sigma(\log(\mathbb{P}_r^\star(\xi_1|s_0)) - \log(\mathbb{P}_r^\star(\xi_2|s_0))). \tag{12}$$

Interestingly, it turns out that solving a soft RL problem like Eq. 10 can be thought of as a *reverse* KL projection (Nielsen, 2020) from $\mathbb{P}_r^\star$ onto the set of policy-induced trajectory distributions.

**Lemma 2.1** (Soft RL is an RKL Projection). *Let $r \in \mathcal{R}$ be a global or local reward model. Define $\pi^\star = \arg\min_{\pi \in \Pi} \mathbb{D}_{KL}(\mathbb{P}_\pi || \mathbb{P}_r^\star)$ as the trajectory-level reverse KL projection of $\mathbb{P}_r^\star$ onto $\Pi$. Next, define $\pi_r^\star$ as the soft-optimal policy computed by solving Eq. 10 over $\Pi$. Then, $\pi^\star = \pi_r^\star$. [Proof]*

We prove a version with a reverse KL regularization to a reference policy in the appendix. In summary, we can view the online two-stage RLHF procedure as *(1)* minimizing the forward KL from the data over reward models in $\mathcal{R}$ to find $\hat{r}_{\text{mle}}$, before *(2)* minimizing the trajectory-level reverse KL over policies in $\Pi$ to the soft-optimal policy under $\hat{r}_{\text{mle}}$. We summarize this argument visually in Fig. 2.

## 2.6 EQUIVALENCES WITH ISOMORPHIC CLASSES

The preceding subsections beg the question: *what does taking a detour through a reward model buy us if we immediately project back to policy space?* As we prove below, under certain assumptions, *all we have done is taken a more circuitous route to likelihood maximization.* We now state our first equivalence result: assuming exact optimization, no reference policy, and most critically, *isomorphic reward and policy classes*, online and offline PFT will produce the same solution:

**Theorem 2.2** (RLHF is MLE when $\mathcal{R} = \mathcal{R}(\Pi)$). *Assume that $\mathcal{R} = \mathcal{R}(\Pi)$ (i.e. they both cover the same set of reward functions regardless of representation). Define $\hat{r}_{\text{mle}}$ as in Eq. 6, $\hat{\pi}_{\text{mle}}$ as in Eq. 9, and $\hat{\pi}_{\text{rlhf}} = \{\arg\max_{\pi \in \Pi} \mathbb{E}_{\xi \sim \pi}[\tilde{r}_{\text{mle}}(\xi)] + \mathbb{H}(\pi) | \tilde{r}_{\text{mle}} \in \hat{r}_{\text{mle}}\}$. Then, we have $\hat{\pi}_{\text{rlhf}} = \hat{\pi}_{\text{mle}}$. [Proof]*

In more traditional terminology, *MLE is invariant to re-parameterization.* Under a *realizability* assumption, we can prove an analogous result with regularization to a reference policy:

**Theorem 2.3** (RLHF is DPO when $\mathcal{R} = \mathcal{R}(\Pi)$). *Assume that $\mathcal{R} = \mathcal{R}(\Pi)$ (i.e. they both cover the same set of reward functions regardless of how they are represented). Define $\hat{r}_{\text{mle}}$ as in Eq. 6 and*

$$\ell_{\text{dpo}}(\pi) = \sum_i^N \log \sigma \left( \sum_h^H \log \frac{\pi(a_{h,i}^+|s_{h,i}^+)}{\pi_{\text{ref}}(a_{h,i}^+|s_{h,i}^+)} - \log \frac{\pi(a_{h,i}^-|s_{h,i}^-)}{\pi_{\text{ref}}(a_{h,i}^-|s_{h,i}^-)} \right).$$

*Next, define $\hat{\pi}_{\text{dpo}} = \arg\max_{\pi \in \Pi} \ell_{\text{dpo}}(\pi)$, $\pi_{\text{dpo}}^\star = \arg\max_{\pi \in \{\mathcal{S} \to \Delta(\mathcal{A})\}} \ell_{\text{dpo}}(\pi)$, $\pi_{\text{mle}}^\star = \arg\max_{\pi \in \{\mathcal{S} \to \Delta(\mathcal{A})\}} \ell_{\text{mle}}(\pi)$, and $\hat{\pi}_{\text{rlhf}} = \{\arg\max_{\pi \in \Pi} \mathbb{E}_{\xi \sim \pi}[\tilde{r}_{\text{mle}}(\xi)] - \mathbb{D}_{KL}(\mathbb{P}_\pi || \mathbb{P}_{\pi_{\text{ref}}}) | \tilde{r}_{\text{mle}} \in \hat{r}_{\text{mle}}\}$. Then, if $\pi_{\text{mle}}^\star, \pi_{\text{dpo}}^\star \subset \Pi$, we have $\hat{\pi}_{\text{rlhf}} = \hat{\pi}_{\text{dpo}}$. [Proof]*

In summary, the preceding results tell us that under certain assumptions, *all roads lead to likelihood* – i.e., expending computation for on-policy sampling provides no discernible benefit over offline MLE. We now attempt to square these idealized theoretical results with the reality of empirical practice.

## 3 ON THE VALUE OF REINFORCEMENT LEARNING IN FINE-TUNING

To better understand where the preceding theory falls short, we now proceed by performing a series of controlled experiments. We focus on the task of learning to summarize from preference feedback (Stiennon et al., 2020) (`tl;dr`) on the `pythia` series of models (Biderman et al., 2023). We report the winrate of our trained models against human-generated references, as evaluated by `gpt-4o`.

**Controlling for Confounders.** To make an apple-to-apples comparison between online and offline PFT methods, we make a concerted effort to hold factors other than on-policy feedback constant. First, to rule out the confounder of a different loss function for offline (e.g., DPO (Rafailov et al.,

2023)) and online (e.g., PPO (Schulman et al., 2017)) PFT, we use *the same DPO loss* (Rafailov et al., 2023) for all training, both because of its practical ubiquity and because there is no clear way to use an RM in offline PFT. [1] Thus, our experiments focus on (*offline*) DPO (as originally proposed) vs. *online* DPO. Second, we follow standard practice and *train global RMs using the same preference data and starting from the same SFT checkpoint as we use for offline DPO*, with a single additional linear layer bolted on for global RMs (Rafailov et al., 2023). Thus, our experiments can be seen as a relatively faithful implementation of the "isomorphic classes" setting we explored theoretically in §2.

For the second stage of online PFT, we perform online DPO (Guo et al., 2024). Concretely, starting from the base policy, we sample 25 completions for each prompt in the preference dataset, rank these completions according to the RM, and use the top and bottom of the list as the preferred and dis-preferred completions for DPO loss minimization. We regularize to whatever policy generated the data. We emphasize that the *only* change between offline and online DPO is the training data – *we use all of the same hyperparameters for all training runs.* Below, we use `Online DPO (Model)` to refer to online DPO using on-policy samples from the policy in parentheses.

**Online DPO > Offline DPO.** In Figure 3, we see that despite our best efforts to control for confounding variables, *we see a significant gap between online and offline DPO*, contradicting §2's theory. While any experiment will have imperfect optimization unlike we assumed in §2, we note it is *equally* imperfect across both online and offline DPO, and therefore unlikely to be the differentiating factor.

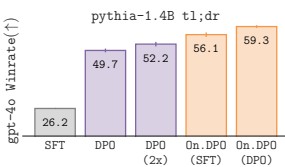 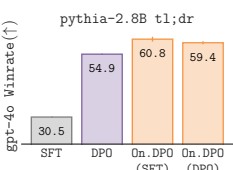

Figure 3: We see online PFT robustly outperform offline PFT on `tl;dr` as judged by `gpt-4o`, despite using the same SFT checkpoint and preference data for training all policies / reward models. We report winrates for all methods across 3 seeds.

Perhaps the most immediate question one might have is whether we are simply running more gradient steps for online DPO than for offline DPO. However, both `DPO` and `Online DPO (SFT)` start from the same SFT model and take the same number of gradient steps, yet we see a difference in performance. The same is true for `DPO(2x)` (in which we perform two epochs of offline DPO) and `Online DPO (DPO)` (in which we perform online DPO starting from the output of offline DPO). Also noteworthy is the improved performance of `Online DPO (DPO)` over the base `DPO` policy: without *any* extra human feedback / data, a global RM trained *on the same preference data* as was used for DPO is able to further improve the `DPO` policy. In some sense, there appears to be more "juice to squeeze" from RMs than from policies, even when both are trained on the same human data.

In short, the results of our preceding carefully controlled experiments echo those of prior work: online PFT robustly outperforms offline PFT (Tajwar et al., 2024; Xu et al., 2024; Song et al., 2024a; Tang et al., 2024; HuggingFace, 2024). We note analogous results have been found in the SFT (Sun & van der Schaar, 2024; Chen et al., 2024; Wulfmeier et al., 2024) and reasoning (Chu et al., 2025; Setlur et al., 2025; Xiang et al., 2025) domains, suggesting a general phenomena beyond just PFT.

---

**Appendix A: A Quintet of Hypotheses for the Online-Offline Gap.**

To explain this performance gap, we explore 6 hypotheses in total. Due to limited space, we highlight only the hypothesis that we failed to falsify, while relegating all the other 5 to App. A. There, we present controlled experiments that rule out alternative explanations for the gap, including $\mathbb{H}_1$: Intrinsic Value of Online Samples (§A.1), $\mathbb{H}_2$: Failure of Offline PFT Regularization to $\pi_{\text{ref}}$ (§A.2), $\mathbb{H}_3$: Relative Ease of Online PFT Optimization (§A.3), $\mathbb{H}_4$: Global RMs Can Be Trained on More Data (§A.4), and $\mathbb{H}_5$: Global RMs Generalize Better OOD (§A.5). To summarize App. A, we observe that despite the lack of information-theoretic separation, online PFT out-performs offline PFT across different sampling distributions, labelers, and model sizes. Furthermore, it appears to be "easier" to learn a global RM than it is to learn a local RM, evidenced by better ID and OOD generalization. Together, our theoretical and empirical results beg the question of whether there is some feature of the problem we are not accounting for in our theory.

---

[1] Note that the choice of the DPO loss rules out explanations that rest on reward model variance (Razin et al., 2025), as no algorithm (either offline or online) gets to see raw reward model outputs, only binarized labels.

## 4 GENERATION-VERIFICATION GAPS IN FINE-TUNING

To set up our final hypothesis $\mathbb{H}_6$, we first note that for many problems of practical interest, the reward function is a simpler object (i.e., can be represented by a circuit of lower depth) than the corresponding (soft) optimal policy. For example, this is the heart of the classical argument for the *inverse RL* approach to imitation (i.e., learning a reward model from demonstrations and decoding it via RL) over behavioral cloning (i.e., directly learning a policy via MLE) given by Ng et al. (2000).

By framing policies as *generators* and reward models as *verifiers*, this argument can be viewed as part of a widespread phenomenon in computer science, where *generation* is often harder than *verification* (Godel, 1956). Indeed, if the widely-held belief that $\mathsf{P} \neq \mathsf{NP}$ (Cook, 2023) is true, there must then exist such problems. For these problems in $\mathsf{NP}$ (loosely, polynomial-time verification) but not $\mathsf{P}$ (loosely, polynomial-time generation), we know there must exist a polynomial-depth verifier circuit but not necessarily a polynomial-depth generator circuit (Cook, 2000). Uniform convergence then tells us that we would need fewer samples to accurately fit a verifier than a generator, as a less expressive function class could be used to approximate the former (Vapnik & Chervonenkis, 2015).

While initially promising, the story becomes more complex when we account for the facts that *(a)* uniform convergence often does not accurately predict the behavior of deep learning (Nagarajan & Kolter, 2019) and *(b)* even if it did, in the isomorphic classes setting (i.e., $\mathcal{R} = \mathcal{R}(\Pi)$), we have the same number of verifiers / generators to search over. This means that we should have the same sample complexity for both MLE problems. Indeed, Ng et al. (2000) implicitly assume that $|\mathcal{R}| \ll |\Pi|$.

However, a wide variety of works have observed that overparameterized models like deep neural networks, when optimized with stochastic gradient descent, learn lower-depth circuits with fewer samples than higher-depth circuits. One can attribute this to either an *implicit regularization* effect of the training algorithm (e.g., the bias of SGD towards minimum norm solutions (Soudry et al., 2024; Gunasekar et al., 2017; Li et al., 2018)), architectural *inductive biases* towards simpler functions (e.g., the double descent phenomenon (Belkin et al., 2019; Nakkiran et al., 2021), the spectral bias of deep networks (Rahaman et al., 2019), or those mentioned in some explanations of grokking (Varma et al., 2023) and length generalization (Zhou et al., 2023)), deep learning performing approximate *Solomonoff Induction* (Schulman, 2023), or certain *data-dependent* complexity measures (Arora et al., 2019). Regardless of one's preferred reasoning as to why, using a larger network than necessary often doesn't hurt sample complexity in practice (Krizhevsky et al., 2012; Radford et al., 2019).

Building on these observations, we now propose a novel hypothesis $\mathbb{H}_6$ for the online-offline performance gap on problems with *(1)* a generation-verification gap and *(2)* a reward model class $\mathcal{R}$ where simpler functions can be learned with fewer samples (e.g., deep neural networks like transformers):

> $\mathbb{H}_6$: **Online PFT is *Proper* Policy Learning.**
>
> *For fine-tuning problems with a simpler underlying reward function than (soft) optimal policy and a reward model class $\mathcal{R}$ that enables learning simpler functions with fewer samples, the first step of online fine-tuning is finding a relatively simple reward model $\hat{r}_{sim} \in \mathcal{R}_{sim} \subset \mathcal{R}$, with the second step then finding a (soft) optimal policy for $\hat{r}_{sim}$, $\hat{\pi}_{rlhf} \in \Pi(\mathcal{R}_{sim})$. Thus, end to end, online fine-tuning only has to choose between policies in $\Pi(\mathcal{R}_{sim}) \subset \Pi$, rather than across all of $\Pi$.*

In essence, the above hypothesis is stating that offline FT solves the harder, *improper* (Valiant, 1984) learning problem over *all* of $\Pi$, while online FT only has to solve the easier *proper* learning problem over $\Pi(\mathcal{R}_{sim}) \subset \Pi$, thereby reducing sample complexity and leading to better policy performance. Put differently, $\mathbb{H}_6$ states that there is a *statistical* separation between online and offline (P)FT. [2]

Under the above hypothesis, if we could somehow constrain offline FT to only pick policies that are (soft) optimal for relatively simple reward models (i.e., $\hat{\pi}_{mle} \in \Pi(\mathcal{R}_{sim})$), we would achieve the same results as online FT. We can prove this under (slightly) weaker assumptions than isomorphic classes:

---

[2]In various theoretical settings, one can show there is a *computational* speedup from improper learning (e.g., Shalev-Shwartz et al. (2012)). However, such a relaxation can incur a significant sample complexity increase if the encompassing class we now have to search over is large enough. Another perspective on $\mathbb{H}_6$ is that the computational-statistical trade-off may not be in favor of improper learning on practical PFT problems.

**Theorem 4.1** (RLHF is MLE Over $\Pi(\mathcal{R}_{sim})$). *Let $\mathcal{R}_{sim} \subset \mathcal{R}$ be an arbitrary subset of the space of reward models and let*

$$\Pi(\mathcal{R}_{sim}) = \left\{ \arg\min_{\pi \in \Pi} \mathbb{D}_{KL}(\mathbb{P}_\pi || \mathbb{P}_r^\star) \middle| r \in \mathcal{R}_{sim} \right\}$$

*be the corresponding set of soft-optimal policies. Also, let*

$$\hat{r}_{sim} = \arg\min_{r \in \mathcal{R}_{sim}} \mathbb{E}_{(\xi_1, \xi_2) \sim \mathcal{D}} \left[ \mathbb{D}_{KL}(\mathbb{P}_\mathcal{D}(\xi_1 \succ \xi_2 | s_0) || \mathbb{P}_r^{BT}(\xi_1 \succ \xi_2 | s_0)) \right],$$

$$\hat{\pi}_{rlhf} = \left\{ \arg\max_{\pi \in \Pi} \mathbb{E}_{\xi \sim \pi}[\tilde{r}_{mle}(\xi)] + \mathbb{H}(\pi) \middle| \tilde{r}_{sim} \in \hat{r}_{sim} \right\},$$

$$\hat{\pi}_{sim} = \arg\min_{\pi \in \Pi(\mathcal{R}_{sim})} \mathbb{E}_{(\xi_1, \xi_2) \sim \mathcal{D}} \left[ \mathbb{D}_{KL}(\mathbb{P}_\mathcal{D}(\xi_1 \succ \xi_2 | s_0) || \mathbb{P}_{r_\pi}^{BT}(\xi_1 \succ \xi_2 | s_0)) \right].$$

*Then, if $\left\{ \pi_r^\star \middle| r \in \mathcal{R}_{sim} \right\} \subset \Pi$, $\hat{\pi}_{rlhf} = \hat{\pi}_{sim}$. [Proof]*

In words, this theorem is saying that if the secondary, RL-based reverse KL projection is without loss, RLHF will recover the MLE over the constrained policy space $\Pi(\mathcal{R}_{sim})$. Critically, it is unclear how else one could actually enforce this constraint other than by the two-stage procedure of first learning a relatively simple RM before optimizing it via RL.[3] Intuitively, we can view this hypothesis as stating that *while all roads lead to likelihood, RL on a simple RM lets us take a shortcut through $\Pi$.*

## 4.1 FAILING TO FALSIFY $\mathbb{H}_6$

First, one may naturally ask what evidence we have for verification being easier than generation for summarization problems. In response, we provide evidence that the underlying reward function is well-approximated by a smaller network than one needs for the policy. In Fig. 4, we observe that using a global RM that is *significantly smaller* than the generation policy leads to nearly identical BoN performance as using an RM that is the same size as the policy. The converse claim is also true: using a global RM that is significantly *larger* than the generation policy leads to no discernible improvement in terms of BoN performance. Taken together, these experiments suggest that the verifier is qualitatively simpler (i.e., well-approximated by a shallower depth circuit) than the generator.

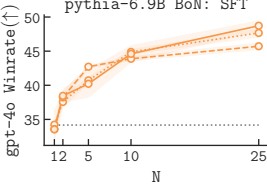 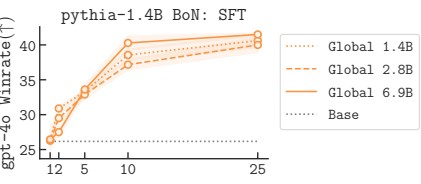

Figure 4: Scaling up reward model size doesn't improve BoN performance, but scaling up policy size does. Results across 3 seeds.

We next observe that $\mathbb{H}_6$ correctly predicts **all** of the experimental results in App. A, which falsified the other hypotheses. In short, *none of these experiments altered the relative complexity of generation versus verification of the underlying problem, which explains why we consistently saw a performance gap.* First, the effective reduction in search space explains the improved performance we saw in Fig. 3. Next, the generation-verification gap is unchanged by prompt augmentation (Fig. 8), different sample and label distributions (Fig. 9), and provides a concrete explanation for the observed difference in in-distribution validation likelihoods between local and global RMs (Fig. 10): global RMs only need to learn a simpler reward function, while local RMs must learn a fundamentally more complex object (equivalent to a generator) to effectively fit the training set. Such better in-distribution margins would likely correlate with the better OOD generalization of global RMs relative to local RMs in Fig. 11.

To further attempt to falsify $\mathbb{H}_6$, we make a testable prediction: *on tasks where there is limited if any generation-verification gap, online PFT would be unlikely to out-perform offline PFT.* For example, if we were to use a reward function for grading the quality of summarization that is as complicated

---

[3]An interesting direction of future work is *off-policy* RL algorithms that can take advantage of learned RMs.

as the corresponding (soft) optimal policy, we would expect the previously consistent gap between online and offline PFT to vanish. We explore 2 ways of eliminating the generation-verification gap:

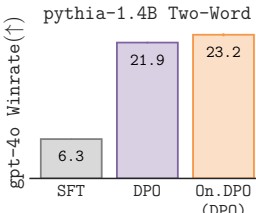
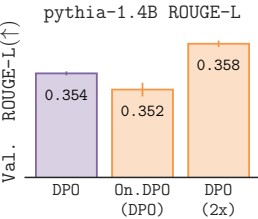

Figure 5: On the bandit-like task of two-word summarization ($\approx \frac{1}{10}$ of the usual horizon $H$), we see online PFT provide a minimal at best improvement of $\approx 1\%$ winrate over offline PFT, unlike *all* prior results. Results across 3 seeds.

Figure 6: When we label data and evaluate completions with the ROUGE-L metric against unobserved reference summaries, we see online PFT provided no improvement over offline PFT, unlike *all* prior results. Results across 3 seeds.

**Closing the Generation-Verification Gap I.** One way to reduce the relative complexity of generation compared to that of verification is to reduce the horizon $H$ of the problem. In the extreme of $H = 1$ (i.e., a contextual bandit (Li et al., 2010)), the complexity of learning the reward function and of learning the (soft) optimal policy should be the same as we can directly compute the latter from the former with a simple action-level / "token-wise" softmax and vice-versa (i.e., no real multi-step planning / reinforcement learning required). Intuitively, setting $H = 1$ is *simplifying the policy.*

We instantiate this idea with the task of generating *two-word* summaries (i.e., less than $\frac{1}{10}$ of the preceding maximum generation length). Given we don't have ground-truth human preferences for this task, we use `gpt-4o` to generate pairs of summaries, rank these summaries, provide reference summaries on the test set of prompts, and compute the final winrate numbers. As before, we use this data for offline DPO as well as for learning the global RM. Continuing to parallel the above, we then sample from the offline DPO policy, rank these completions with the global RM, and use this dataset for an iteration of online DPO. In Figure 5, we see that in contrast to all prior experiments that kept the complexity of generation higher than that of verification, *we see that online DPO does not meaningfully improve the performance of the offline DPO policy, as predicted by* $\mathbb{H}_6$.

**Closing the Generation-Verification Gap II.** Another way to eliminate the gap is by choosing a reward function from which we can easily "read off" the (soft) optimal policy without an explicit planning / reinforcement learning procedure. Specifically, we conduct an experiment using the ROUGE-L metric (Lin, 2004), which essentially counts how many words of the generated summary appear (in order) in an unobserved human-generated reference summary, as a reward function. For such a problem, the minimal-complexity verifier would need to include a lookup table that maps from prompts to the exact text of the unobserved reference summaries, a tall order. Furthermore, given this lookup table, it would be trivial to generate optimally, implying that generation and verification should be of similar complexities. Intuitively, this setup is *complicating the reward function.*

To create our preference dataset, we sample from the policy and use the ROUGE-L metric to rank samples, as well as for evaluation. In Figure 6, *we see that doing an iteration of online DPO with feedback from a learned global RM does not improve the performance of the base offline DPO policy,* unlike all but the previous result, as predicted by $\mathbb{H}_6$. Doing an extra epoch of offline DPO does (slightly) improve the ROUGE-L score, implying offline DPO has not reached optimal performance.

We conclude this section by noting that similar results on the benefits of learning a verifier rather than directly learning a generator for fine-tuning have been observed in both the SFT (Sun & van der Schaar, 2024; Choudhury, 2025) – where this flavor of approach is more commonly known as *inverse RL* (Ziebart, 2010; Swamy et al., 2021) – and reasoning domains (Chu et al., 2025; Setlur et al., 2025; Xiang et al., 2025). These results can be seen as further evidence for $\mathbb{H}_6$, outside of the PFT domain.

## 4.2 Isomorphisms Aren't Always a Two-Way Street

A lingering question in the reader's mind might be "*if policies and rewards are isomorphic in soft RL, why does it take more data to learn the former?*" In particular, we know from Eq. 11 that we

can map from rewards to trajectory distributions, from which we can extract a policy via soft value iteration (Ziebart, 2010). Perhaps the most impressive insight of Rafailov et al. (2023) is that we can *invert* this isomorphism (up to the partition function $Z(r, s_0)$ which cancels out in the Bradley-Terry likelihood) and instead express a local reward model in terms of its implied soft-optimal policy, as in Eq. 9. However, as later observed by Rafailov et al. (2024), it is more accurate to think of a local RM as a *Q-function* rather than as a reward function: a fundamentally different and often more complex object that needs to perform token-level credit assignment and encode multi-step reasoning.

In greater detail, we know from Ziebart (2010) that $\pi_r^\star$ has a (soft) $Q$-function given by

$$Q_r^\star(s_h, a_h) = \log \left( \sum_{\xi \in \Xi | s_h, a_h} \exp(r(\xi)) \right), \pi_r^\star(a_h | s_h) = \frac{\exp(Q_r^\star(s_h, a_h))}{\sum_{a \in \mathcal{A}} \exp(Q_r^\star(s_h, a))} = \frac{\exp(Q_r^\star(s_h, a_h))}{V_r^\star(s_h)}.$$

Observe that the next-token logits of a softmax policy are its implied soft $Q$-function (up to a state-wise shift). Also, recall that these were the optimization variables in our offline MLE problem (Eq. 9). Thus, MLE over softmax policies (i.e., fitting local RMs) corresponds to directly fitting $Q$-functions.

Defining $\mathcal{Q}_\mathcal{R}^\star = \{Q_r^\star | r \in \mathcal{R}\}$ as the set of soft-Q functions for some $r \in \mathcal{R}$, we know that $\mathcal{R} \cong \mathcal{Q}_\mathcal{R}^\star$ by construction. However, *just because such an isomorphism exists, it does not need to be true that both endpoints are equally easy to represent:* one endpoint (e.g., $\mathcal{Q}_\mathcal{R}^\star$) could require *far* deeper circuits to be well-approximated than the other (e.g., $\mathcal{R}$). When learning algorithms optimize over circuit classes that contain these function classes, uniform convergence would then tell us we need more samples to learn the former than the latter, as we have to search over circuits of greater depth. Thus, even though $\mathcal{R} \cong \mathcal{Q}_\mathcal{R}^\star$, the fact that we are actually fitting these functions from data means that we have to pay statistically for their relative representational complexities during offline MLE.

To build some intuition for the complexity of $r$ vs. $Q_r^\star$, let us consider navigating through a maze with a single goal with $+1$ reward. Then, a circuit that represents $r$ only needs to encode the position of the goal, while a circuit that represents $Q_r^\star$ needs to encode the *path* to the goal from *any* position in the maze. As we scale up the horizon $H$ of the planning problem, the optimal policy $\pi_r^\star$ can reach the goal from a wider set of states. Thus, $Q_r^\star$ becomes more complex, as nonzero value is achieved on a broader swathe of the state space. As $\pi_r^\star$ is a simple softmax over $Q_r^\star$, it also grows in complexity.

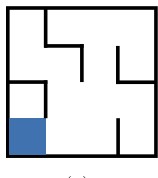 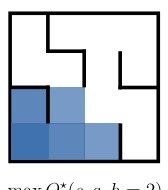 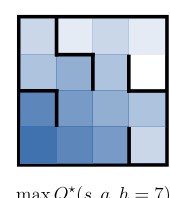

$r(s)$     $\max_{a \in \mathcal{A}} Q_r^\star(s, a, h = 1)$     $\max_{a \in \mathcal{A}} Q_r^\star(s, a, h = 2)$     $\max_{a \in \mathcal{A}} Q_r^\star(s, a, h = 7)$

Figure 7: Consider maze navigation. Reward $r$ just needs to encode the goal, while $Q_r^\star$ needs to encode the distance to the goal from anywhere. $Q_r^\star$ grows more complex with $h$.

As we discussed in §4, such a difference in circuit complexity between $r$ and $\pi_r^\star$ or $Q_r^\star$ is one way of formalizing the notion of a generation-verification gap. This means that beyond the examples we explore in §4, App. A, and Fig. 7's maze example, for problems in NP but not P, we should expect that $\pi_r^\star$ and $Q_r^\star$ require a deeper circuit to represent than $r$. Then, a learning algorithm that doesn't have the ability to search over just $\mathcal{Q}_\mathcal{R}^\star$ (e.g., via RL on all $r \in \mathcal{R}$) must instead search over a class of *deeper* circuits containing $\mathcal{Q}_\mathcal{R}^\star$. Thus, learning $Q_r^\star$ or $\pi_r^\star$ requires more data than to learn $r$.

Intuitively, these RL problems in NP but not P require meaningful, multi-step reasoning to solve. However, not all RL problems require multi-step reasoning / have a generation-verification gap. In fact, in §4.1, we provide multiple examples of such problems for which the greedy ($H = 1$) policy is optimal and don't see a difference in terms of the number of samples required to learn $r$ vs. $\pi_r^\star$ or $Q_r^\star$. These problems have an *effective horizon* (Laidlaw et al., 2023) of $H \approx 1$. Thus, $r$, $\pi_r^\star$, and $Q_r^\star$ have similar circuit complexities and require similar amounts of data to learn, matching our results.

To close, one might counter that whether we first fit a reward function and then apply RL to it or directly fit a $Q$ function, we eventually want to end up with a policy. While true, recall that the second stage of online PFT requires no more preference data samples: we only pay in terms of computation. Thus, a different perspective on $\mathbb{H}_6$ is that online PFT *trades data for computation*: rather than paying *statistically* for directly fitting a potentially more complex $Q$ function via MLE like offline PFT, online PFT instead pays *computationally* during the secondary RL procedure. One can view this as a *computational-statistical trade-off* (Chandrasekaran & Jordan, 2013) for preference fine-tuning.

In summary, *while isomorphisms go both ways, they aren't always a two-way street*: the two endpoints of an isomorphism can have dramatically different circuit complexities, which can manifest as a statistical cost when fitting the more complex endpoint (e.g., $\mathcal{Q}_{\mathcal{R}}^{\star}$). Instead, one can fit the simpler endpoint (e.g., $\mathcal{R}$) and compute the isomorphism (e.g., via RL) to save data at the cost of compute.

## 5 DISCUSSION

Practically, $\mathbb{H}_6$ tells us that for problems where we believe verification is simpler than generation, it is a better use of limited human preference data to learn verifiers rather than generators. Hypothesis $\mathbb{H}_6$ also suggests that for increasingly complex problems that require longer-range *planning* (e.g. multi-turn RLHF, agentic tasks, or even real-world robotics), we should see an even greater separation between online and offline PFT than we saw in our experiments. $\mathbb{H}_6$ also suggests variety of future research directions. For example, one might ensure that current reward model architectures are able to accurately represent human preferences. As argued by Swamy et al. (2024), the diversity of rater views often results in *intransitive preferences*, which can't be rationalized by *any* reward model. More broadly, given our overarching framing in terms of likelihood maximization, analogous arguments to ours for $\mathbb{H}_6$ could be made beyond the PFT setting (e.g., for the SFT / imitation learning setting).

## 6 ACKNOWLEDGMENTS

We thank Geoff Gordon and Kiante Brantley for helpful discussions, and Zhaolin Gao and Yuda Song for experimental advice. GKS and ZSW were supported in part by a STTR grant. WS acknowledges funding from NSF IIS-2154711, NSF CAREER 2339395, and DARPA LANCER: LeArning Network CybERagents. SC is supported in part by a Google Faculty Research Award, OpenAI SuperAlignment Grant, ONR Young Investigator Award, NSF RI #2312956, and NSF FRR#2327973.

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

# A   A QUINTET OF HYPOTHESES FOR THE ONLINE-OFFLINE GAP

CONTENTS

We now consider (and in some cases, falsify) a series of hypotheses, both from prior work and of our own, that attempt to explain the discrepancy between §2's theory and the reality of PFT in §3.

## A.1   $\mathbb{H}_1$: INTRINSIC VALUE OF ONLINE SAMPLES.

Intuitively, it may seem as though getting feedback on samples from the policy is providing qualitatively new information from the fixed offline dataset. Tang et al. (2024) come to this conclusion.

However, it is not clear as to what, precisely, is the mechanism by which this on-policy data actually helps with policy optimization when we account for the fact that the labels for this data are merely *imputed* by a RM trained on the *same* off-policy dataset, rather than truly new information from some ground-truth reward function. Put differently, there is no information in the RM that wasn't either in preference dataset or the (shared) base model, both of which the offline methods have access to.

Recall that from an information-theoretic perspective, we know on-policy data is redundant via the data-processing inequality (MacKay, 2003) – we cannot create any new information (i.e., *bona fide* human preferences) to learn from via sampling from the policy. Auto-regressive generation also doesn't get new information from a real environment. Furthermore, given the RM, one could in theory run a (soft) value iteration procedure to compute the optimal policy without ever sampling on-policy, meaning that *on-policy samples are technically not necessary to solve the (soft) RL problem.*

## A.2   $\mathbb{H}_2$: FAILURE OF OFFLINE PFT REGULARIZATION TO $\pi_{\mathsf{REF}}$ .

Via an elegant argument, Song et al. (2024a) prove that offline PFT algorithms like DPO (Rafailov et al., 2023) and IPO (Azar et al., 2023) require global (i.e., stronger) preference dataset *coverage* conditions than online PFT methods, as they are unable to effectively regularize to the reference policy otherwise. Intuitively, this is because reverse KL regularization involves an expectation over samples drawn from the learner's policy. Thus, all generable trajectories need to be within the support of the preference dataset for a purely offline algorithm to effectively control the reverse KL. A variety of approaches, from sampling on-policy to directly calculate the reverse KL and adding it in as an auxiliary loss term (Song et al., 2024a), to several forms of offline pessimism (Huang et al., 2024b; Cen et al., 2024; Fisch et al., 2024; Liu et al., 2024) have been proposed to mitigate this issue.

While certainly a contributing factor, there are several pieces of evidence that imply this distinction does not explain all of the gap in performance between online and offline approaches to PFT. First, adding in a reverse KL penalty to DPO doesn't completely close the gap to *bona fide* online PFT methods (Song et al., 2024a; Gao et al., 2024a). Second, PFT methods that don't explicitly regularize to the prior like SimPO (Meng et al., 2024) have shown strong performance on multiple benchmarks. Third, there are fine-tuning problems where staying close to the reference policy is not particularly important for performance, and we still see online methods out-perform offline methods (Choudhury & Sodhi, 2024). Lastly, *all the experiments in Figure 3 use the same regularizer for both online and offline algorithms, and we still observe a gap in performance.* Thus, $\mathbb{H}_2$ fails to predict these results.

## A.3   $\mathbb{H}_3$: RELATIVE EASE OF ONLINE PFT OPTIMIZATION.

One might ask if offline PFT is somehow faced with a harder optimization problem than online PFT because the former is forced to escape extra local minima, as argued by Xu et al. (2024).

However, as we remarked above, because we use the same loss function (DPO's) for both offline and online PFT, the only difference between the procedures we're comparing is the data we're passing in. It is unclear how the *same* number of online samples could make it easier to optimize the *same* loss function – Xu et al. (2024) instead compare offline DPO to online PPO (Schulman et al., 2017).

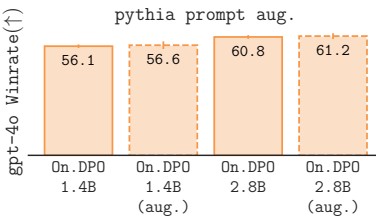

Figure 8: Augmenting the set of prompts we use for online DPO with those from the SFT dataset ($\approx$ 3x as much data) barely improves winrate. This is contrary to what a refinement of $\mathbb{H}_3$ predicts. Results across 3 training seeds.

A refinement of the above hypothesis is grounded in the phenomenon of *computational-statistical gaps:* for certain problems, additional samples, even if information-theoretically redundant, reduce the amount of computation required to find a solution (Daniely et al., 2013; Bandeira et al., 2018). For example, filling in more of a Sudoku puzzle makes it easier to solve, even if we've already observed the minimum set of numbers to uniquely specify the solution. One might therefore view the (redundant) on-policy samples as providing these extra "constraints" on the policy search space.

To attempt to falsify this refined hypothesis, we perform *prompt augmentation* to increase the size of the preference dataset $\mathcal{D}$ we use for training the online DPO policy. Specifically, we generate on-policy and compute RM labels on prompts from the SFT dataset, nearly tripling the training set size. Intuitively, if the refined hypothesis were true, we'd expect that we would see an increase in policy performance with these "redundant" samples. However, in Figure 8, *we instead observe almost no improvement in downstream winrate*, which the refined version of $\mathbb{H}_3$ fails to predict.

### A.4 $\mathbb{H}_4$: GLOBAL RMs CAN BE TRAINED ON MORE DATA.

When we look at the most performant global reward models available (Zhu et al., 2023; Wang et al., 2024; Lambert et al., 2024), we often observe that they are trained on a relatively wide distribution of data compared to the preference datasets used for offline PFT. It is therefore a natural question as to whether there is something intrinsic about global RMs that make them more amenable to training on a wider distribution of data than local RMs / policies.[4] In fact, the human preference data we use to train our global RMs and policies is generated by a diverse assortment of comparisons from a wide variety of policies (Stiennon et al., 2020), rather than just samples from our particular base policy.

To attempt to falsify this hypothesis, we generate a more concentrated dataset by using samples *only* from the SFT policy, with winners picked by `gpt-4o` query. As we are reducing the diversity of the preference dataset, $\mathbb{H}_4$ predicts that the gap between online and offline PFT would shrink.

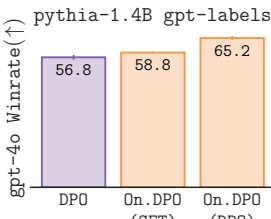

Figure 9: Performing an iteration of online DPO (third bar) on top of offline DPO (first bar) significantly improves winrate, even when all samples in $\mathcal{D}$ are generated by the SFT policy and labeled by `gpt-4o`, a narrower preference dataset. This is contrary to what $\mathbb{H}_4$ predicts. All results reported across 3 seeds.

In Figure 9, *we see online DPO (third bar) on top of offline DPO (first bar) still significantly improving performance, even though we train all models with a relatively narrow, on-policy dataset*, which $\mathbb{H}_4$ fails to predict. We note that it is unsurprising that online DPO on top of the SFT policy (second bar) roughly matches the performance of offline DPO (first bar) – the former procedure amounts to merely relabeling the same SFT policy samples with the RM instead of the ground-truth `gpt-4o` labels.

---

[4]There is a $\mathcal{O}(H)$ computational speedup provided by not needing to do a per-token forward + backward pass for global RMs as in local RM training, which could allow more data to be trained on with the same computation budget. However, we use the same amount of data for local/global RM training in *all* experiments.

Thus, while it is hard to argue that more (and more diverse) data wouldn't lead to a better RM, we don't see evidence that policies and RMs would have different levels of efficacy of taking advantage of this wider distribution of data, at least for the sorts of problems we consider in our work.

## A.5   $\mathbb{H}_5$: GLOBAL RMS GENERALIZE BETTER OOD.

In their comprehensive empirical study, Tajwar et al. (2024) argue that when the peak of the learned reward model falls outside of the support of the preference data, online PFT techniques are better able to maximize this reward than offline approaches. Implicitly, such explanations are assuming that reward models generalize better OOD than policies. Lin et al. (2024) observe that, empirically, the reward model implied by DPO generalizes less well OOD than a standard, global reward model, a perspective echoed by the top of the RewardBench leaderboard (Lambert et al., 2024). The work of Chu et al. (2025) finds evidence of this phenomenon in vision-based reasoning tasks.

However, there are several open questions in the above story. First, given persistent concerns about reward model over-optimization (Gao et al., 2023; Eisenstein et al., 2023), it seems that neither policies nor RMs generalize all that well OOD in general. Second, it is not immediately clear why policies and reward models that are trained from the *same* initial checkpoint on the *same* dataset (as in the experiments in Fig. 3) should generalize differently OOD. Third, the comparison between DPO RMs and standard global RMs changes two factors at once: DPO regularizes to the reference policy (unlike unregularized global reward model training) and optimizes a local (rather than a global) RM.

We conduct a several experiments to provide answers to the preceding questions. To remove the confounder of the reference, we also train local RMs without regularization (i.e., reference-less DPO), which we refer to as `Local`. We note that this is precisely the $\hat{\pi}_{\mathsf{mle}}$ (Eq. 9) we analyzed above.

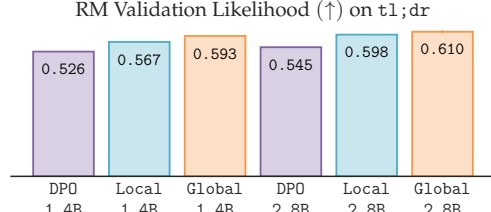

Figure 10: Going from a global RM to a local RM or from a local RM to a DPO RM hampers (in-distribution) validation accuracy. We report results for all RMs across 3 seeds.

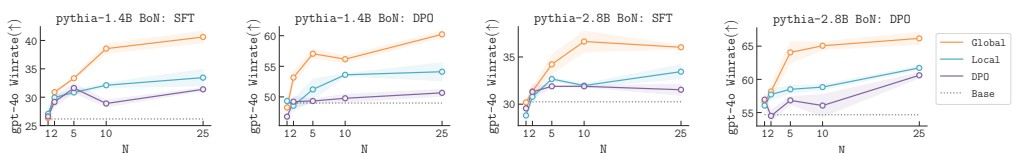

Figure 11: We compare global, local, and DPO reward models in terms of their BoN winrate as a measure of OOD generalization. We use the same size for the policy and the reward model and generate samples either from a base SFT or offline DPO policy. As we increase $N$, we see a perfect correlation emerge between the in-distribution validation likelihood reported in Fig. 10 and BoN performance. All standard errors are reported across three training seeds.

We first compare DPO, local, and global RMs of the same backbone in terms of their *validation likelihood* (i.e., in terms of their *in-distribution* generalization when evaluated as classifiers). In Figure 10, we consistently see that enforcing a token-wise decomposition leads to worse performance in distribution. Furthermore, adding in regularization during training further hampers the hold-out validation likelihood of the local RM. These results persist across multiple model sizes.

We next evaluate the *OOD generalization* (as in, not on samples from the same distribution as the human preference dataset) of each of these families of models by evaluating their Best-Of-$N$ (BoN) performance on samples from both the SFT and offline DPO policies. Higher values of $N$ correspond

to testing a reward model on a wider swathe of generations. In Figure 11, we see higher in-distribution validation likelihood *perfectly* correlate with BoN performance for high enough $N$. [5]

Better in-distribution margins are known to correlate with some kinds of OOD generalization (Miller et al., 2021). Intuitively, by having a larger margin in-distribution, one hopes a classifier will be able to better separate OOD samples, similar to classical arguments for SVMs. While this observation provides a potential explanation for the better OOD performance reported in prior work, it merely kicks the can down the road: *we still haven't explained why there should be a difference in-distribution in the first place.* Thus, while we don't falsify $\mathbb{H}_5$, the root cause of the observed phenomena of global reward models generalizing better than local reward models is still unclear, leaving a conceptual gap.

---

[5]The results where we sample from a DPO policy and further filter the samples with the internal DPO RM and improve winrate are separately interesting as they provide evidence of *self-improvement*, echoing the findings of Song et al. (2024b); Huang et al. (2024a). However, we note that an external global RM is still significantly better at accurately ranking learner samples across the board.

# B    PROOFS.

## CONTENTS

## B.1    PROOF OF LEMMA 2.1

*Proof.*

$$
\begin{aligned}
\pi^\star = \arg\min_{\pi\in\Pi} \mathbb{D}_{\mathrm{KL}}(\mathbb{P}_\pi||\mathbb{P}_r^\star) &= \arg\min_{\pi\in\Pi} \mathbb{E}_{s_0\sim\rho_0}\left[\sum_{\xi\in\Xi|s_0}\mathbb{P}_\pi(\xi|s_0)(\log(\mathbb{P}_\pi(\xi|s_0)) - \log(\mathbb{P}_r^\star(\xi)))\right] \\
&= \arg\min_{\pi\in\Pi} \mathbb{E}_{s_0\sim\rho_0}\left[\sum_{\xi\in\Xi|s_0}\mathbb{P}_\pi(\xi|s_0)(\log(\mathbb{P}_\pi(\xi|s_0)) - r(\xi) + \log Z(r,s_0))\right] \\
&= \arg\min_{\pi\in\Pi} \mathbb{E}_{s_0\sim\rho_0}\left[\sum_{\xi\in\Xi|s_0}\mathbb{P}_\pi(\xi|s_0)(\log(\mathbb{P}_\pi(\xi)) - r(\xi))\right] \\
&= \arg\max_{\pi\in\Pi} \mathbb{E}_{s_0\sim\rho_0}\left[\sum_{\xi\in\Xi|s_0}\mathbb{P}_\pi(\xi|s_0)(-\log(\mathbb{P}_\pi(\xi|s_0)) + r(\xi))\right] \\
&= \arg\max_{\pi\in\Pi} \mathbb{E}_{\xi\sim\pi}[r(\xi)] + \mathbb{H}(\pi) \\
&= \pi_r^\star.
\end{aligned}
$$

$\square$

### B.1.1    PROOF OF LEMMA 2.1 WITH REFERENCE POLICY

Given two distributions $\mathbb{P}, \mathbb{Q} \in \Delta(\Xi)$, define their trajectory-level mixture $\mathbb{P} \cdot \mathbb{Q}$ as

$$
(\mathbb{P} \cdot \mathbb{Q})(\xi) = \frac{\mathbb{P}(\xi)\mathbb{Q}(\xi)}{\sum_{\xi'\in\Xi}\mathbb{P}(\xi')\mathbb{Q}(\xi')}, \forall \xi \in \Xi. \tag{13}
$$

**Lemma B.1** (MinRelEnt RL is an RKL Projection). *Let $r \in \mathcal{R}$ be a global reward model. Define $\pi^\star = \arg\min_{\pi\in\Pi} \mathbb{D}_{KL}(\mathbb{P}_\pi||\mathbb{P}_r^\star \cdot \mathbb{P}_{\pi_{ref}})$ as the trajectory-level reverse KL projection of the trajectory-level mixture of $\mathbb{P}_r^\star$ and $\mathbb{P}_{\pi_{ref}}$ onto $\Pi$. Next, define $\pi_r^\star$ as the soft-optimal policy computed by solving*

$$
\pi_r^\star = \arg\max_{\pi\in\Pi} \mathbb{E}_{\xi\sim\pi}[r(\xi)] - \mathbb{D}_{KL}(\mathbb{P}_\pi||\mathbb{P}_{\pi_{ref}}). \tag{14}
$$

*Then, $\pi_r^\star = \pi^\star$.*

*Proof.*

$$\pi^\star = \arg\min_{\pi \in \Pi} \mathbb{D}_{\mathrm{KL}}(\mathbb{P}_\pi || \mathbb{P}_r^\star \cdot \mathbb{P}_{\pi_{\mathrm{ref}}})$$

$$= \arg\min_{\pi \in \Pi} \mathbb{E}_{s_0 \sim \rho_0} \left[ \sum_{\xi \in \Xi | s_0} \mathbb{P}_\pi(\xi | s_0)(\log(\mathbb{P}_\pi(\xi | s_0)) - \log(\mathbb{P}_r^\star(\xi)) - \log(\mathbb{P}_{\pi_{\mathrm{ref}}}(\xi | s_0))) \right]$$

$$= \arg\min_{\pi \in \Pi} \mathbb{E}_{s_0 \sim \rho_0} \left[ \sum_{\xi \in \Xi | s_0} \mathbb{P}_\pi(\xi | s_0)(\log(\mathbb{P}_\pi(\xi | s_0)) - r(\xi) + \log Z(r, s_0) - \log(\mathbb{P}_{\pi_{\mathrm{ref}}}(\xi | s_0))) \right]$$

$$= \arg\min_{\pi \in \Pi} \mathbb{E}_{s_0 \sim \rho_0} \left[ \sum_{\xi \in \Xi | s_0} \mathbb{P}_\pi(\xi | s_0)(\log(\mathbb{P}_\pi(\xi)) - r(\xi) - \log(\mathbb{P}_{\pi_{\mathrm{ref}}}(\xi | s_0))) \right]$$

$$= \arg\max_{\pi \in \Pi} \mathbb{E}_{s_0 \sim \rho_0} \left[ \sum_{\xi \in \Xi | s_0} \mathbb{P}_\pi(\xi | s_0)(-\log(\mathbb{P}_\pi(\xi | s_0)) + r(\xi) + \log(\mathbb{P}_{\pi_{\mathrm{ref}}}(\xi | s_0))) \right]$$

$$= \arg\max_{\pi \in \Pi} \mathbb{E}_{\xi \sim \pi}[r(\xi)] - \mathbb{D}_{\mathrm{KL}}(\mathbb{P}_\pi || \mathbb{P}_{\pi_{\mathrm{ref}}})$$

$$= \pi_r^\star.$$

$\square$

## B.2 PROOF OF THEOREM 2.2

*Proof.* Below, we use $\tilde{\pi}_{\mathsf{mle}} \in \hat{\pi}_{\mathsf{mle}}$ and $\tilde{r}_{\mathsf{mle}} \in \hat{r}_{\mathsf{mle}}$ to refer to specific minima of Eqs. 9 and 6.

First, we observe that because $\mathcal{R} = \mathcal{R}(\Pi)$ and the fact that minimizing the same functional over the same function class must produce the same set of minima, $\hat{r}_{\mathsf{mle}} = \mathcal{R}(\hat{\pi}_{\mathsf{mle}})$. This further implies that for each $\tilde{\pi}_{\mathsf{mle}} \in \hat{\pi}_{\mathsf{mle}}$, $\exists \tilde{r}_{\mathsf{mle}} \in \hat{r}_{\mathsf{mle}}$ such that $\forall \xi \in \Xi$, $r_{\tilde{\pi}_{\mathsf{mle}}}(\xi) = \tilde{r}_{\mathsf{mle}}(\xi)$ and vice-versa.

From Lemma 2.1, we know that each $\tilde{\pi}_{\mathsf{rlhf}} \in \hat{\pi}_{\mathsf{rlhf}}$ can also be written as the minimizer of a reverse KL projection:

$$\tilde{\pi}_{\mathsf{rlhf}} = \arg\min_{\pi \in \Pi} \mathbb{D}_{\mathrm{KL}}(\mathbb{P}_\pi || \mathbb{P}_{\tilde{r}_{\mathsf{mle}}}^\star)$$

$$= \arg\min_{\pi \in \Pi} \mathbb{D}_{\mathrm{KL}}(\mathbb{P}_\pi || \mathbb{P}_{r_{\tilde{\pi}_{\mathsf{mle}}}}^\star)$$

$$= \arg\min_{\pi \in \Pi} \mathbb{D}_{\mathrm{KL}}(\mathbb{P}_\pi || \mathbb{P}_{\tilde{\pi}_{\mathsf{mle}}})$$

$$= \tilde{\pi}_{\mathsf{mle}},$$

where the last step uses the fact that $\tilde{\pi}_{\mathsf{mle}} \in \Pi$. We can repeat the above steps for each $\tilde{r}_{\mathsf{mle}} \in \hat{r}_{\mathsf{mle}}$ to prove that $\hat{\pi}_{\mathsf{rlhf}} = \hat{\pi}_{\mathsf{mle}}$. $\square$

## B.3 PROOF OF THEOREM 2.3

*Proof.* Below, we use $\tilde{\pi}_{\mathsf{mle}} \in \hat{\pi}_{\mathsf{mle}}$ and $\tilde{r}_{\mathsf{mle}} \in \hat{r}_{\mathsf{mle}}$ to refer to specific minima of Eqs. 9 and 6.

First, we observe that because $\mathcal{R} = \mathcal{R}(\Pi)$ and the fact that minimizing the same functional over the same function class must produce the same set of minima, $\hat{r}_{\mathsf{mle}} = \mathcal{R}(\hat{\pi}_{\mathsf{mle}})$. This further implies that for each $\tilde{\pi}_{\mathsf{mle}} \in \hat{\pi}_{\mathsf{mle}}$, $\exists \tilde{r}_{\mathsf{mle}} \in \hat{r}_{\mathsf{mle}}$ such that $\forall \xi \in \Xi$, $r_{\tilde{\pi}_{\mathsf{mle}}}(\xi) = \tilde{r}_{\mathsf{mle}}(\xi)$ and vice-versa.

Next, observe that for all datasets $\mathcal{D}$ and $\forall \tilde{\pi}_{\mathsf{dpo}}^\star \in \pi_{\mathsf{dpo}}^\star$, $\exists \tilde{\pi}_{\mathsf{mle}}^\star \in \pi_{\mathsf{mle}}^\star$ such that $\forall \xi \in \Xi$,

$$\sum_h^H \log \tilde{\pi}_{\mathsf{dpo}}^\star(a_h | s_h) = \sum_h^H \log \tilde{\pi}_{\mathsf{mle}}^\star(a_h | s_h) + \log \pi_{\mathsf{ref}}(a_h | s_h). \tag{15}$$

Next, because of our realizability assumptions (i.e. $\pi_{\mathsf{mle}}^\star, \pi_{\mathsf{dpo}}^\star \subset \Pi$), we also know that $\hat{\pi}_{\mathsf{mle}} = \pi_{\mathsf{mle}}^\star$ and $\hat{\pi}_{\mathsf{dpo}} = \pi_{\mathsf{dpo}}^\star$. Put together, this tell us that $\forall \, \tilde{\pi}_{\mathsf{dpo}} \in \hat{\pi}_{\mathsf{dpo}}, \exists \tilde{\pi}_{\mathsf{mle}} \in \hat{\pi}_{\mathsf{mle}}$ such that $\forall \xi \in \Xi$,

$$\sum_h^H \log \tilde{\pi}_{\mathsf{dpo}}(a_h|s_h) = \sum_h^H \log \tilde{\pi}_{\mathsf{mle}}(a_h|s_h) + \log \pi_{\mathsf{ref}}(a_h|s_h). \tag{16}$$

Then, substituting $\tilde{r}_{\mathsf{mle}}$ for the equivalent $r_{\tilde{\pi}_{\mathsf{mle}}}$, we arrive at

$$\sum_h^H \log \tilde{\pi}_{\mathsf{dpo}}(a_h|s_h) = \tilde{r}_{\mathsf{mle}}(\xi) + \sum_h^H \log \pi_{\mathsf{ref}}(a_h|s_h). \tag{17}$$

This tells us that $r_{\tilde{\pi}_{\mathsf{dpo}}} = \tilde{r}_{\mathsf{mle}} + r_{\tilde{\pi}_{\mathsf{ref}}}$, and therefore $\mathbb{P}^\star_{r_{\tilde{\pi}_{\mathsf{dpo}}}} = \mathbb{P}^\star_{\tilde{r}_{\mathsf{mle}}} \cdot \mathbb{P}_{\pi_{\mathsf{ref}}}$. Next, from Lemma B.1, we know that each $\tilde{\pi}_{\mathsf{rlhf}} \in \hat{\pi}_{\mathsf{rlhf}}$ can also be written as the minimizer of a reverse KL projection:

$$\tilde{\pi}_{\mathsf{rlhf}} = \arg \min_{\pi \in \Pi} \mathbb{D}_{\mathsf{KL}}(\mathbb{P}_\pi || \mathbb{P}^\star_{\tilde{r}_{\mathsf{mle}}} \cdot \mathbb{P}_{\pi_{\mathsf{ref}}}) \tag{18}$$

$$= \arg \min_{\pi \in \Pi} \mathbb{D}_{\mathsf{KL}}(\mathbb{P}_\pi || \mathbb{P}^\star_{r_{\tilde{\pi}_{\mathsf{dpo}}}}) \tag{19}$$

$$= \arg \min_{\pi \in \Pi} \mathbb{D}_{\mathsf{KL}}(\mathbb{P}_\pi || \mathbb{P}_{\tilde{\pi}_{\mathsf{dpo}}}) \tag{20}$$

$$= \tilde{\pi}_{\mathsf{dpo}}, \tag{21}$$

where the last step uses the fact that $\tilde{\pi}_{\mathsf{dpo}} \in \Pi$. We can repeat the above steps for each $\tilde{r}_{\mathsf{mle}} \in \hat{r}_{\mathsf{mle}}$ to prove that $\hat{\pi}_{\mathsf{rlhf}} = \hat{\pi}_{\mathsf{dpo}}$. □

## B.4 PROOF OF THEOREM 4.1

*Proof.* Below, we use $\tilde{\pi}_{\mathsf{sim}} \in \hat{\pi}_{\mathsf{sim}}$ and $\tilde{\pi}_{\mathsf{rlhf}} \in \hat{\pi}_{\mathsf{rlhf}}$ to refer to specific minima.

First, we observe that because of Lemma 2.1, we can write that

$$\hat{\pi}_{\mathsf{rlhf}} = \left\{ \arg \min_{\pi \in \Pi} \mathbb{D}_{\mathsf{KL}}(\mathbb{P}_\pi || \mathbb{P}^\star_{\tilde{r}_{\mathsf{sim}}}) \, \middle| \, \tilde{r}_{\mathsf{sim}} \in \hat{r}_{\mathsf{sim}} \right\}. \tag{22}$$

We can then combine the above with the fact that $\hat{r}_{\mathsf{sim}} \subset \mathcal{R}_{\mathsf{sim}}$ to conclude that $\hat{\pi}_{\mathsf{rlhf}} \subset \Pi(\mathcal{R}_{\mathsf{sim}})$.

Now, we assume for the sake of contradiction that $\exists \tilde{\pi}_{\mathsf{rlhf}} \in \hat{\pi}_{\mathsf{rlhf}}$ such that $\tilde{\pi}_{\mathsf{rlhf}} \notin \hat{\pi}_{\mathsf{sim}}$ (i.e., $\tilde{\pi}_{\mathsf{rlhf}}$ is not a BT likelihood maximizer over $\Pi(\mathcal{R}_{\mathsf{sim}})$). Then, by the fact that $\tilde{\pi}_{\mathsf{rlhf}} \in \hat{\pi}_{\mathsf{rlhf}} \subset \Pi(\mathcal{R}_{\mathsf{sim}})$ and the definition of each $\tilde{\pi}_{\mathsf{sim}} \in \hat{\pi}_{\mathsf{sim}}$ as a BT likelihood maximizer over $\Pi(\mathcal{R}_{\mathsf{sim}})$, it must be true that

$$\mathbb{E}_{(\xi_1,\xi_2)\sim\mathcal{D}}\left[\mathbb{D}_{\mathsf{KL}}(\mathbb{P}_{\mathcal{D}}(\xi_1 \succ \xi_2|s_0)||\mathbb{P}^{\mathsf{BT}}_{r_{\tilde{\pi}_{\mathsf{sim}}}}(\xi_1 \succ \xi_2|s_0))\right]$$
$$< \mathbb{E}_{(\xi_1,\xi_2)\sim\mathcal{D}}\left[\mathbb{D}_{\mathsf{KL}}(\mathbb{P}_{\mathcal{D}}(\xi_1 \succ \xi_2|s_0)||\mathbb{P}^{\mathsf{BT}}_{r_{\tilde{\pi}_{\mathsf{rlhf}}}}(\xi_1 \succ \xi_2|s_0))\right].$$

Then, because we assumed that $\forall \tilde{r}_{\mathsf{sim}} \in \hat{r}_{\mathsf{sim}} \subset \mathcal{R}_{\mathsf{sim}}, \pi^\star_{\tilde{r}_{\mathsf{sim}}} \in \Pi$ (i.e., the secondary RKL projection is exact), it must then be true that $\exists \tilde{r}_{\mathsf{sim}} \in \hat{r}_{\mathsf{sim}}$ s.t. $\tilde{\pi}_{\mathsf{rlhf}} = \pi^\star_{\tilde{r}_{\mathsf{sim}}}$, which further implies that

$$\mathbb{E}_{(\xi_1,\xi_2)\sim\mathcal{D}}\left[\mathbb{D}_{\mathsf{KL}}(\mathbb{P}_{\mathcal{D}}(\xi_1 \succ \xi_2|s_0)||\mathbb{P}^{\mathsf{BT}}_{r_{\tilde{\pi}_{\mathsf{rlhf}}}}(\xi_1 \succ \xi_2|s_0))\right]$$
$$= \mathbb{E}_{(\xi_1,\xi_2)\sim\mathcal{D}}\left[\mathbb{D}_{\mathsf{KL}}(\mathbb{P}_{\mathcal{D}}(\xi_1 \succ \xi_2|s_0)||\mathbb{P}^{\mathsf{BT}}_{r_{\pi^\star_{\tilde{r}_{\mathsf{sim}}}}}(\xi_1 \succ \xi_2|s_0))\right]$$
$$= \mathbb{E}_{(\xi_1,\xi_2)\sim\mathcal{D}}\left[\mathbb{D}_{\mathsf{KL}}(\mathbb{P}_{\mathcal{D}}(\xi_1 \succ \xi_2|s_0)||\mathbb{P}^{\mathsf{BT}}_{\tilde{r}_{\mathsf{sim}}}(\xi_1 \succ \xi_2|s_0))\right].$$

Together, these imply that:

$$\mathbb{E}_{(\xi_1,\xi_2)\sim\mathcal{D}}\left[\mathbb{D}_{\mathsf{KL}}(\mathbb{P}_{\mathcal{D}}(\xi_1 \succ \xi_2|s_0)||\mathbb{P}^{\mathsf{BT}}_{r_{\tilde{\pi}_{\mathsf{sim}}}}(\xi_1 \succ \xi_2|s_0))\right]$$
$$< \mathbb{E}_{(\xi_1,\xi_2)\sim\mathcal{D}}\left[\mathbb{D}_{\mathsf{KL}}(\mathbb{P}_{\mathcal{D}}(\xi_1 \succ \xi_2|s_0)||\mathbb{P}^{\mathsf{BT}}_{\tilde{r}_{\mathsf{sim}}}(\xi_1 \succ \xi_2|s_0))\right].$$

However, by our realizability assumption, we also know that $\forall \tilde{\pi}_{\mathsf{sim}} \in \hat{\pi}_{\mathsf{sim}} \subset \Pi(\mathcal{R}_{\mathsf{sim}}), r_{\tilde{\pi}_{\mathsf{sim}}} \in \mathcal{R}_{\mathsf{sim}}$. Thus, this expression contradicts the definition of $\tilde{r}_{\mathsf{sim}}$ as a BT likelihood maximizer over $\mathcal{R}_{\mathsf{sim}}$. □

In words, the realizability assumption of the above theorem is that the policy class $\Pi$ includes all policies that are soft-optimal for the relatively simple verifiers in $\mathcal{R}_{\mathsf{sim}}$, but not necessarily for all verifiers in $\mathcal{R}$. Thus, *fully* isomorphic classes implies but is not implied by this (weaker) assumption.

# C  EXPERIMENTAL DETAILS.

CONTENTS

Throughout all of our experiments, we use the `tl;dr` data from https://github.com/openai/summarize-from-feedback, optimize models from the `pythia` family (Biderman et al., 2023), and build upon the codebase for REBEL (Gao et al., 2024a), available at https://github.com/ZhaolinGao/REBEL. We use VLLM for fast inference (Kwon et al., 2023). We will open-source the code, models, and data for this project upon paper acceptance.

## C.1  DATASET DETAILS

For the standard summarization tasks (Figs. 3, 8, 10, 11, 4), our training data has the following characteristics:

Table 1: Dataset split, prompts, and maximum generation length for *TL;DR* summarization

| DATASET | TRAIN/VAL/TEST | PROMPT | GENERATION LENGTH |
|---------|----------------|--------|-------------------|
| SFT | 117K/6.45K/6.55K | "TL;DR:" | 53 |
| PFT | 92.9K/83.8K/- | "TL;DR:" | 53 |

As is standard practice, we first fine-tune the base `pythia` models on the SFT training set before performing some variant of DPO on the PFT training set.

For the prompt augmentation results in Figure 8, we also use the prompts from the SFT training set for on-policy generation.

For the "GPT-label" results in Figure 9, we sample twice from the SFT policy with temperature 0.1 and use the same prompting strategy we use for evaluation to pick a winner with `gpt-4o`. We perform this procedure all PFT training prompts to generate a new training set.

For the two-word summarization results in Figure 5, we reduce the maximum generation length to 5 tokens as well as replacing "TL;DR:" with "Two-Word TL;DR:". We use the following prompt to generate a pair of two-word summaries via `gpt-4o` query:

```
Given the following forum post, please write a summary of the post
    of length at most two words. The summaries should not include
    any unimportant or irrelevant details.

### Post:
{{post}}

### Instructions:
Your response should use the format:
Summary: <two-word summary>

Do not include any other text or explanations in your response.
```

We use the same prompting strategy as for evaluation to pick a winner. We do this for all prompts in the PFT training set.

For the ROUGE-L results in Figure 6, we rank 25 prompt completions sampled from the SFT policy with temperature 0.1 on each of the PFT training prompts via the ROUGE-L F1 score against the

preferred prompt completion in the training set. We use the highest and lowest scoring generations from the SFT policy as the preferred / dis-preferred completions.

### C.1.1 ONLINE DPO DATASET DETAILS

The above discussion is on how we get the preference dataset we use for training reward models and offline DPO policies. For online DPO, we sample from either the SFT or Offline DPO policy 25 times with temperature 0.1, rank the samples according to some kind of reward model, and use the top and bottom of the list as the preferred and dis-preferred completions to the prompt for a new training set. Also, unlike some implementations of online DPO (e.g. those in Gao et al. (2024a)), we do not continuously sample from the policy and instead sample once from the initial policy across all prompts in a "batched" fashion to reduce computational expense. This further aligns our online and offline PFT implementations.

### C.2 MODEL DETAILS

Across all experiments in this paper, we use the `pythia` series of models. In particular, we use the 1.4B, 2.8B, and 6.9B parameter de-duped variants available at https://huggingface.co/collections/EleutherAI/pythia-scaling-suite-64fb5dfa8c21ebb3db7ad2e1 as our base models. We remove the final softmax and add in a single linear layer to transform a policy model into a global reward model. We train all policies and reward models ourselves and do not use LoRA anywhere. We train SFT models via 1 epoch over the SFT dataset. We the further fine-tune these models via one epoch of training on the PFT data to create local and global reward models. We also only perform one epoch of training for DPO on the PFT data, except for the results marked as `DPO (2x)` in Figs. 3 and 6. We use a mixture of A100s, A800s, and H100s across experiments. No training run took more than 8 hours.

### C.3 TRAINING DETAILS

We only run three training algorithms in this paper: logistic regression for global reward model training, SFT and online/offline DPO. We use consistent hyperparameters for both of these procedures across *all* experiments in this paper. We use AdamW (Loshchilov et al., 2017) for all optimization in this paper.

| PARAMETER | VALUE |
|---|---|
| BATCH SIZE | 64 |
| LEARNING RATE | 3E-6 |
| $\varepsilon$ | 1E-5 |
| SCHEDULE | COSINE DECAY |

Table 2: Hyperparameters for global RM training.

| PARAMETER | VALUE |
|---|---|
| BATCH SIZE | 128 |
| LEARNING RATE | **3E-7** |
| $\varepsilon$ | 1E-5 |
| $\beta$ | 0.05 |
| SCHEDULE | LINEAR DECAY |

Table 3: Hyperparameters for DPO training. We found a relatively low learning rate lead to significantly better performance for DPO across the board. For offline DPO, we set $\pi_{\text{ref}} = \pi_{\text{sft}}$. For online DPO, we set $\pi_{\text{ref}}$ to whatever generated the training data (i.e. either $\pi_{\text{sft}}$ or $\pi_{\text{dpo}}$ depending on the experiment) and scale logits by the inverse of the sampling temperature $\frac{1}{0.1} = 10$ before taking the softmax, as is standard practice.

| PARAMETER | VALUE |
|---|---|
| $\beta$ | $\frac{1}{H} = \frac{1}{53}$ |

Table 4: Hyperparameters for local reward model training that differ from those for DPO. There is no $\pi_{\text{ref}}$ for local RMs.

## C.4 EVALUATION DETAILS

We sample with temperature 0.01 for all winrate / ROUGE-L policy evaluations. We use temperature 0.1 for all BoN results.

For all winrate computations, we use `gpt-4o`. For standard summarization, we use the following prompt:

```
Which of the following summaries does a better job of summarizing
    the most important points in the given forum post, without
    including unimportant or irrelevant details? Judge based on
    accuracy, coverage, and coherence.

### Post:
{{post}}

### Summary A:
{{summarya}}

### Summary B:
{{summaryb}}

### Instructions:
FIRST provide a one-sentence comparison of the two summaries,
    explaining which \
you prefer and why. SECOND, on a new line, state only "A" or "B"
    to indicate your choice. Your response should use the format:
Comparison: <one-sentence comparison and explanation>
Preferred: <"A" or "B">
```

We use 600 randomly sampled prompts from the SFT test set for evaluation.

For two-word summarization (i.e. Fig 5) we use the following prompt:

```
Which of the following two-word summaries does a better job of
    summarizing the most important points in the given forum post,
     without including unimportant or irrelevant details? Judge
    based on accuracy, coverage, and coherence.

### Post:
{{post}}

### Summary A:
{{summarya}}

### Summary B:
{{summaryb}}

### Instructions:
FIRST provide a one-sentence comparison of the two summaries,
    explaining which \
you prefer and why. SECOND, on a new line, state only "A" or "B"
    to indicate your choice. Your response should use the format:
```

```
Comparison: <one-sentence comparison and explanation>
Preferred: <"A" or "B">
```

We use 6000 randomly sampled prompts from the SFT test set for evaluation.

For ROUGE-L evaluations (i.e. Fig. 6), we use the *F1* variant of the metric against preferred completions from the PFT validation set.

For likelihood evaluations, we use the appropriate validation loss for the model class. For global reward models, this is just logistic loss. For DPO reward models, this includes the scaling factor $\beta$ and the reference policy probabilities. For local reward models, this includes just the scaling factor $\beta$.

