# OpenReview forum: "All Roads Lead to Likelihood: The Value of Reinforcement Learning in Fine-Tuning"
_ICLR.cc/2026/Conference — ICLR 2026 Poster_

### Official Review · Reviewer_J9QF · 2025-10-28

**Soundness:** 2
**Presentation:** 3
**Contribution:** 3
**Rating:** 6
**Confidence:** 3

**Summary:**

This paper discusses several hypotheses for the benefit of online fine-turning in preference-based RL, focussing on large language models.   Starting from a unified objectives for online and offline fine-tuning, it shows that both online and offline fine-tuning can be seen as maximizing data likelihood, although for the online setting the learned policies is within a restricted policy class induced by learned reward models. Then, this paper raises a new assumption for the benefit of online fine-tuning: simpler reward models induce smaller policy classes, which turns out to have better performance. Finally, the paper offers evidence for the benefit of using simpler reward models.

**Strengths:**

1. Overall, the presentation of this paper is easy to follow.

2. The discussion through the lens of maximum likelihood estimation is suitable for the LLM setting and offers new insight for understanding the benefit of online fine-tuning.

3. In particular, the assumption of the connection between simper reward models and better performance is interesting and worth investigation. For example, verifiable rewards used for training reasoning models are also simple ones.

**Weaknesses:**

1. The presentation of H6, which is the core assumption made in this paper, is hard to understand in the first glimpse. I would say that $\Pi(R_\text{sim})\subset \Pi$ is a general statement, since $R_\text{sim}$ is unlikely to cover **all possible reward functions**. A better way for stating the benefit of using small reward models would strengthen this paper. Alternatively, you can define $R$ to be all the reward function consisting with the preferences, so that $\Pi$ will be all the policies that can generate the preferences.

2. The empirical justification is very limited. Since the theoretical justifications in this paper are straight-forward and rely on idea conditions, it is better to include more empirical results.

**Questions:**

1.  beside the summary task being considered, is there any other task that also has generation-verification gap?

2. Is the assumption H6 valid for tasks where the generation content is longer than the prompt?

3. Why do you use BoN in Fig.4?

---

> ### Author Response · Authors · 2025-11-20
> **Re:**
>
> We thank the reviewer for their appreciation of our presentation and novel insights. Responding to the weaknesses raised and questions asked:
>
> W1: We’d like to clarify that both $\Pi$ and $\mathcal{R}$ are function classes (e.g., transformers of some size) rather than the set of all policies / rewards. We also don’t think it is generically true that $\Pi(\mathcal{R}_{\text{sim}}) \subset \Pi$ – one could imagine having a much more expressive class of reward models than policies. Similarly, given how noisy human feedback is, it is unlikely there is any reward function in our class that is perfectly “consistent” with the preference data. This is why we frame our paper in terms of likelihood maximization, which is always possible.
>
> W2: We include extensive experiments to falsify other hypotheses and support H6 in Appendix A.
>
> Q1: Yes! As we mention in Lines 319-325, all tasks in NP but not P provably have this property. This includes a wide variety of problems from satisfiability, to scheduling, to logical reasoning problems like Sudoku. More intuitively, we believe the majority of tasks where we collect human preference labels rather than human generations have this flavor, as if generation was as easy as verification, we’d simply ask the person to do the task for us and learn via imitation.
>
> Q2: Yes! Our hypothesis doesn’t make any assumptions about the prompt length whatsoever – we instead argue that as the underlying complexity (e.g., task horizon / generation length) increases, the difficulty of verification scales less quickly than the difficulty of generation, and therefore it takes less human feedback data to learn a verifier than a generator.
>
> Q3: We use BoN performance as a measure of the quality of an RM that isn’t tied to a particular policy optimization algorithm.
>
> Please let us know if you have any other questions or concerns and thank-you for engaging in the review process!

---

### Official Review · Reviewer_sSEo · 2025-10-29

**Soundness:** 3
**Presentation:** 2
**Contribution:** 3
**Rating:** 6
**Confidence:** 3

**Summary:**

This paper investigates why two-stage fine-tuning (FT) procedures—training a reward model (RM) followed by reinforcement learning (RL)—often outperform direct offline optimization, despite appearing information-theoretically inefficient.  They propose the hypothesis that such a phenomenon is due to the generation-verification gap.

**Strengths:**

This paper appears to be the first to investigate the underlying reason why the two-stage fine-tuning procedure outperforms purely offline approaches. The proposed hypothesis is well-motivated, and the authors provide a theoretical analysis to support it. Furthermore, they conduct extensive numerical experiments that not only reinforce their hypothesis but also help rule out alternative explanations.

**Weaknesses:**

Overall, I do not have major concerns about this article. The core idea is clearly articulated, and the presentation is well-structured and easy to follow. My only reservation is whether the contribution is substantial enough for publication at ICLR. Although the paper spans nine pages, a significant portion is devoted to setup and related work, with the main theoretical contribution centered on Theorem 3.1.  I would suggest that some proof sketches and key arguments be included in the main text.

**Questions:**

See above.

**Details Of Ethics Concerns:**

N/A.

---

> ### Author Response · Authors · 2025-11-20
> **Re:**
>
> We thank the reviewer for their appreciation of our theoretical and empirical work and the effort we spent ruling out alternative explanations. Responding to the weakness raised:
>
> W1: We agree that including more content in the main paper would strengthen our manuscript. We will include a more detailed discussion / take-aways in our next revision. If there are any parts of the appendix that you would like us to try and squeeze into the main paper, please let us know.
>
> Please let us know if you have any other questions or concerns and thank-you for engaging in the review process!

---

> > ### Comment · Reviewer_sSEo · 2025-11-26
> >
> > I thank the authors for the reply, and I will keep my positive rating for this paper.

---

### Official Review · Reviewer_Ypdj · 2025-10-31

**Soundness:** 3
**Presentation:** 4
**Contribution:** 3
**Rating:** 6
**Confidence:** 3

**Summary:**

This paper tackles the question of why do complex, two-stage online methods (like RLHF) empirically outperform simpler, direct offline methods (like DPO), even when they both optimize the same likelihood-based objective?

The paper's first key finding is theoretical: when the policy and reward model function classes are isomorphic (i.e., $\mathcal{R} = \mathcal{R}(\Pi)$), the optimal solutions for both online and offline methods are identical (Theorems 2.2 & 2.3). This theoretical equivalence contradicts robust empirical findings.

The paper then systematically conducts controlled experiments to rule out several common hypotheses and propose a more suitable hypothesis: the generation-verification gap.

**Strengths:**

The generation-verification gap is a nice conceptual contribution. It reframes the online vs. offline debate to a root cause a "statistical efficiency" problem.


The paper is a model of good scientific reasoning. It cleanly formalises the theory-practice gap (Theorems 2.2/2.3) and then treats various explanations (e.g., optimization, regularization, OOD) as falsifiable hypotheses.

The paper's best evidence comes from its two "gap-closing" experiments. Predicting that the online PFT advantage would disappear on a bandit-like task (Fig 5) and a ROUGE-L task (Fig 6) is not obvious.

The authors online DPO setup -- where an RM is used to re-label on-policy data, which is then fed into the same DPO loss is a clever way to isolate the core variable. It successfully controls for confounders like the optimization algorithm (PPO vs. DPO loss), ensuring the comparison is truly about the two-stage process versus the one-stage process.

**Weaknesses:**

1) The "Simplicity" of Verification is a Black Box: The entire argument of H-6 hinges on the assertion that a verifier is simpler than a generator. This central concept of simplicity is never formally defined.

2) The paper frames H-6 as the sole surviving explanation. This seems unlikely to be true. It's more plausible that other factors are also true and compound the effect. For example, for H-5  (OOD Generalization), Fig 9, Fig 10 show that global RMs do generalize better (both in-distribution and OOD) than local RMs. The paper claims H-6 causes this. But it's equally plausible that better OOD generalization is a separate benefit of the global RM architecture, which then adds to the statistical efficiency benefit of H-6.

3) How does the paper conceptually differ from [1] which seem to propose a similar argument?

4) The initial theoretical question is built on the isomorphism between policies and reward functions. But in practice offline DPO uses a local RM (architecturally tied to the policy, $r_{\pi} = \sum \log \pi$) and online RLHF uses a global RM (using the final hidden state of the full sequence). Does this create a confounder?


[1] Self-Improvement in Language Models: The Sharpening Mechanism

**Questions:**

See weaknesses

---

> ### Author Response · Authors · 2025-11-20
> **Re:**
>
> We thank the reviewer for their appreciation of our theoretical and empirical contributions and our efforts to proceed scientifically when testing both our final hypothesis and alternative explanations. Responding to the weaknesses raised:
>
> W1: In our paper, we take the perspective of circuit complexity, where shallower circuits are simpler. We mention this in the first few paragraphs of Sec. 3.1, note this notion of simplicity agrees with our empirical results above Fig. 4, and mention more precisely characterizing the circuit complexity of rewards and policies as a promising direction of future work in our discussion section.
>
> W2: We agree that we cannot, based on the experiments in our paper, rule out all other potential causes of the generation gap between local and global RMs. That said, given these differ architecturally by a single linear layer, we find it unlikely that this is the dominant factor rather than the simplicity of the underlying function to be learned.
>
> W3: We’re big fans of this paper (and hence cite it in Footnote 5) but believe they are making a fundamentally orthogonal argument. Huang et al. argue that re-ranking greedily decoded generations based on the total log probability of the response can improve performance without external reward feedback. In contrast, we argue that given human preferences, the most sample-efficient method to learn from them is to learn a global reward model, rather than directly learn a policy via DPO.
>
> W4: In our experiments, we do our best to make the local and global RM learning procedures as close to each other as possible. We use the same architecture (except for a single extra linear layer for the global RM), start from the same SFT checkpoint, and use the same preference dataset, for training both. We also use similar logistic regression loss functions for both, ablating away the reference probability in the DPO loss function in the appendix as a differentiating factor. So, while it is hard to say there is no other plausible difference between global and local RMs, we would argue we’ve made a good-faith attempt to control for as many confounders as possible.
>
> Please let us know if you have any other questions or concerns and thank-you for engaging in the review process!

---

### Official Review · Reviewer_Caqs · 2025-11-01

**Soundness:** 3
**Presentation:** 3
**Contribution:** 3
**Rating:** 8
**Confidence:** 4

**Summary:**

This paper aims to provide an intuitive explanation for the empirically observed performance gap between two-stage (e.g., RLHF, online DPO) and direct (e.g., DPO, offline DPO) preference fine-tuning methods. Through a combination of theoretical analysis and empirical studies, the authors evaluate multiple hypotheses concerning the value of reinforcement learning in two-stage approaches. Among the tested hypotheses, only the generation–verification gap (it is easier to learn a simple reward model from preference data and optimize policies for simple verifiers) remained supported by the evidence. The paper concludes that two-stage methods effectively operate over a reduced policy space, requiring less data than offline fine-tuning.

**Strengths:**

The paper is clearly written, and the results are rigorous.

It makes both theoretical and empirical contributions to understanding the origins of the performance gap between online and offline preference fine-tuning.
- shows that when policy and reward classes match, online and offline methods share the same set of optima.
- falsifies several existing hypotheses about the benefits of RL in preference fine-tuning and introduces the generation-verification gap as a plausible alternative explanation.

Provides useful practical insights:
- for problems where verification is simpler than generation, two-stage methods are preferable.
- for tasks requiring long-horizon reasoning or complex planning, the gap between online and offline approaches is likely to widen.

(Note: I have not carefully checked the additional hypotheses in the appendix.)

**Weaknesses:**

Potential related work: Nika et al. (2024) conducted a comparative theoretical analysis of RLHF and DPO, highlighting the generation-verification gap as a key factor explaining when RLHF statistically outperforms DPO. They show that when the reward class is of lower complexity than the policy class, RLHF tends to perform better, consistent with this paper's findings.

Nika et al., 2024. Reward Model Learning vs. Direct Policy Optimization: A Comparative Analysis of Learning from Human Preferences. ICML, 2024.

**Questions:**

Clarification question:

The experiments supporting the generation-verification gap hypothesis show:
- cases where verification is easier than generation, and online methods outperform offline ones;
- cases where complex reward functions diminish the benefit of online techniques.

In LLM preference fine-tuning, even when reward and policy classes have comparable complexity, do we still observe a gap favoring online methods? If that is the case, is the generation-verification gap alone sufficient to explain this phenomenon, or am I missing something?

---

> ### Author Response · Authors · 2025-11-20
> **Re:**
>
> We thank the reviewer for their appreciation of our writing, theoretical and empirical contributions, and practical implications. Responding to the weaknesses and questions raised:
>
> W1: We will include this reference in our next revision.
>
> Q1:
> > In LLM preference fine-tuning, even when reward and policy classes have comparable complexity, do we still observe a gap favoring online methods?
>
> Yes! This is one of the key differences between our work and the above reference. We first observe that modern deep networks often learn simpler functions with less data (see lines 331-341 for potential explanations). This means that even when we have roughly isomorphic policy and reward classes (i.e., the situation we’re in for LLM PFT), learning a simpler verifier likely takes less data (as supported by our experiments in the main paper / appendix). Thus, H6 goes beyond traditional uniform convergence style analyses and takes advantage of the fact that common reward model classes “enable learning simpler functions with fewer samples.” Please let us know if this makes sense or if any extra clarification would be helpful.
>
> Please let us know if you have any other questions or concerns and thank-you for engaging in the review process!

---

> > ### Comment · Reviewer_Caqs · 2025-11-26
> >
> > I thank the authors for addressing my concerns!

---

### Meta-Review · Area_Chair_swbu · 2025-12-16

**Summary:**

The reviewers requested additional citations and explanations of the differences from the classical uniform convergence argument; explicitly define "complexity" as circuit depth in the main text, compare it with another paper on self-improvement, and list the controlled confounding factors; move key content from the appendix into the main text and add a discussion summary; explain that the generation-verification gap corresponds to an NP-not-P problem, clarify that the results are independent of the cue length, explain that the BoN experiment is only used to evaluate the quality of the reward model in isolation, and point out that a certain structural assumption is not universally true, with supplementary experiments attached.

It is worth mentioning that more diverse experiments could be considered to further validate the proposed method.

**Reviewer Concerns:**

The reviewers' comments have been largely addressed.

**Reviewer Scores:**

The reviewers consistently maintained positive scores.

---

### Decision · Program_Chairs · 2026-01-26

Accept (Poster)